# ESTABLISHING KNOWLEDGE PREFERENCE IN LANGUAGE MODELS

## ABSTRACT

Language models are known to encode a great amount of factual knowledge through pretraining. However, such knowledge might be insufficient to cater to user requests, requiring the model to integrate external knowledge sources and adhere to user-provided specifications. When answering questions about ongoing events, the model should use recent news articles to update its response; when asked to provide recommendations, the model should prioritize user specifications over retrieved product reviews; when some facts are edited in the model, the updated facts should override all prior knowledge learned by the model even if they are conflicting. In all of the cases above, the model faces a decision between its own parametric knowledge, (retrieved) contextual knowledge, and user instruction knowledge. In this paper, we (1) unify such settings into the problem of *knowledge preference* and define a three-level preference hierarchy over these knowledge sources; (2) compile a collection of existing datasets IfQA, MQuAKE, and MRQA covering a combination of settings (with/without user specifications, with/without context documents) to systematically evaluate how well models obey the intended knowledge preference; and (3) propose a dataset synthesis method that composes diverse question-answer pairs with user assumptions and related context to directly fine-tune LMs for instilling the hierarchy of knowledge. We demonstrate that a 7B model, fine-tuned on only a few thousand examples automatically generated by our proposed method, effectively achieves superior performance (more than 18% improvement across all evaluation benchmarks) in adhering to the desired knowledge preference hierarchy.

## 1 INTRODUCTION

Language models memorize factual knowledge during pretraining, which allows them to perform open-domain question answering with remarkable accuracy. However, the knowledge encoded within the model (parametric knowledge) might be erroneous or incomplete, falling short of users' expectations. Some applications require the language model to leverage the most recent knowledge, such as the latest election results, or stock prices. This is typically set up as closed-domain QA or retrieval-augmented generation (RAG) where the newer knowledge is presented as extra context to the language model. While much effort has been spent on improving retrieval and ranking results, it would be futile if the model simply disregards the input and sticks to its own "prior beliefs" (Longpre et al., 2021; Yu & Ji, 2023). Even if the model only occasionally appears obstinate, this will largely undermine user trust as now users would need to fact-check every claim against the provided context. In these applications, it is critical to ensure that contextual knowledge is preferred over the models' parametric knowledge. Another type of application including personalized search and recommendation requires the integration of user preferences. User preferences should always be respected over model parametric knowledge and contextual knowledge. Model editing (Meng et al., 2022a;b; De Cao et al., 2021; Mitchell et al., 2022; Zhong et al., 2023) can be seen as a special case of such preferences, where the new facts override learned facts even if they are counterfactual in nature. In all of these settings (RAG, closed-domain QA, integrating user beliefs and model editing), we observe that the key is to enforce a certain priority among knowledge from different sources.

The strife between parametric knowledge and contextual knowledge has been measured across many models and forms of contexts (Longpre et al., 2021; Neeman et al., 2023; Li et al., 2023; Xie et al., 2024; Kortukov et al., 2024). While earlier models (T5 (Raffel et al., 2020), Roberta (Liu et al.,

2019)) seem to be baffled by conflicting knowledge and often stick to their priors (Longpre et al., 2021), recent larger models (OPT (Zhang et al., 2022), GPT-3 (Brown et al., 2020)) show potential in successfully updating their answers through in-context edits (Zheng et al., 2023; Zhong et al., 2023; Si et al., 2023; Kortukov et al., 2024). Existing studies also reveal some influence factors for in-context update failures, such as incoherence context (Xie et al., 2024) and parametric answers (the answer according to parametric knowledge) appearing in context (Kortukov et al., 2024). Under the RAG setting, attempts have been made to rectify model behavior in the presence of noisy retrieval (Zhang et al., 2024a; Yoran et al., 2024), requiring the model to cite retrieved contextual knowledge only when it is relevant to the question. While these lines of work are seemingly separate, we believe that they are just shapes and forms of the same underlying question: *how should language models behave when faced with multiple sources of (noisy) knowledge?*

To answer this question, we first build our framework of hierarchical knowledge preference over three distinct levels: parametric knowledge, contextual knowledge and instruction knowledge. While the divide between parametric and contextual knowledge is not new, we make the further distinction between (retrieved) contextual knowledge and (user or system-provided) instruction knowledge to account for the case of noisy context. This three-level hierarchy unifies multiple settings: (1) prioritizing instruction knowledge over parametric knowledge is the problem of in-context knowledge editing (Zheng et al., 2023); (2) prioritizing contextual knowledge over parametric knowledge is the problem of RAG and closed-domain QA (Zhang et al., 2024a; Yoran et al., 2024); (3) the full hierarchy supports personalized or counterfactual QA with RAG (Yu et al., 2023).

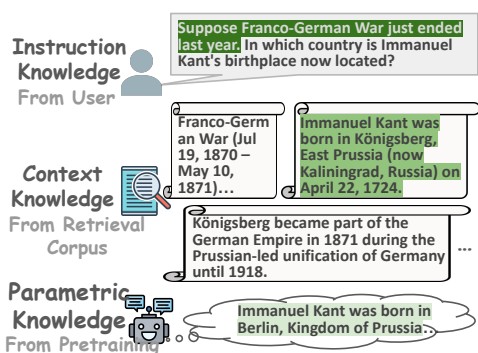

Figure 1: Examples of instruction knowledge, context knowledge and parametric knowledge. Conflicted parts are highlighted. The conflict between instruction knowledge and context knowledge lies in the conflicted timestamps. The conflict between context knowledge and parametric knowledge lies in the conflicted factual knowledge.

To systematically evaluate a model's ability to adhere to the desired knowledge preference hierarchy, we create a benchmark adapted from several existing datasets (IfQA (Yu et al., 2023), MQuAKE (Zhong et al., 2023) and MRQA (Fisch et al., 2019)) to cover all of the aforementioned settings. Moreover, we stress-test the model's behavior in more difficult cases where the contextual knowledge is noisy and the question requires (multi-hop) reasoning. We observe that while large, proprietary models such as GPT-4o can perform relatively well (86.46% $F_1$ on the counterfactual knowledge editing task), open-source models, especially those fine-tuned with open instruction data (Mistral with Alpaca tuning only achieves 28.48% $F_1$ on same task), fail to model this knowledge hierarchy even when they are explicitly instructed to do so in the prompt.

To close this gap, we design a dataset synthesis procedure to create instruction-tuning data that follows our desired order of knowledge preference. We start from Wikipedia and Wikidata, which are known as high-quality sources of factual data, and use GPT-4o to synthesize questions and counterfactual evidence. For multi-hop questions, we sample fact chains from Wikidata, alter some of the intermediate facts, and then synthesize passages to support each hop. Our dataset creation process does not rely on any human annotation and through experiments, we show that a few thousand examples are sufficient to unlock the knowledge preference ability of open-source LLMs (28.48% $F_1$ $\rightarrow$ 89.36% $F_1$ on the counterfactual knowledge editing task without specific prompting). Our model is also more robust when encountering noisy knowledge and shows even more gains on complex, multi-hop questions.

To conclude, our main contributions include:

- We formulate the *knowledge preference* problem of LLMs, which unifies settings where LMs need to decide among parametric knowledge, contextual knowledge, and user instruction knowledge.
- We compile a benchmark to evaluate the knowledge preference property of LMs by adapting existing datasets to cover all combinations of different settings and difficulties. We encourage model

developers to take *knowledge preference* as an additional axis of evaluation as many important applications (RAG, knowledge editing, and user preference modeling) entail this ability.

• We design a data synthesis procedure to automatically create instruction-tuning data for instilling the knowledge preference. We show that fine-tuning an open-source LM with a few thousand dedicated data samples can make the model much more receptive to user instruction knowledge and contextual knowledge, achieving superior performance on all settings in our benchmark.

## 2 FORMULATION OF KNOWLEDGE PREFERENCE

When the parametric knowledge (intrinsic knowledge) (Petroni et al., 2019; Mallen et al., 2022) of an LLM is insufficient to give the correct answer to user queries, we can introduce external knowledge either in the instruction or as additional context.

**Instruction Knowledge** is the knowledge injected through user instructions. Instruction knowledge can refer to rules or principles that govern how the model should utilize other types of knowledge, i.e. problem-solving constraints from user instructions and assumptions from hypothetical questions.

**Context Knowledge** is the potentially noisy context provided to the LLM during inference time. One typical case is the retrieved passages in retrieval-augmented generation. The retrieved passages can provide newly-updated knowledge or domain-specific knowledge which is generally expected to override or complement LLMs' own knowledge in RAG.

We take the RAG case in Fig. 1 as an example where the user queries the LLM with a question (ignore the question assumption first). Resolving the question requires solving a model preference problem where we want the LLM to prioritize relevant knowledge in the retrieved context over knowledge embedded in the LLM's parameters. Sometimes, users will give their own constraints or requirements for answering the query (e.g., the question assumption in Fig. 1). Correspondingly, to fulfill the user requirements, the LLM should override the original way it utilizes the knowledge, by following a new reasoning flow and utilizing different pieces of context knowledge and parametric knowledge. Then, the RAG case in Fig. 1 is fundamentally a knowledge preference problem where we further give the instruction knowledge the highest priority in the inference process. More generally, in this work, we define *Hierarchical Knowledge Preference* built on these types of knowledge.

**Hierarchical Knowledge Preference.** In applications of LLMs, conflicts between instruction knowledge, context knowledge, and parametric knowledge are frequently inevitable. For instance, a user may provide counterfactual hypothesis or unprecedented constraints which may conflict with the retrieved documents or the LLMs' own knowledge (Yu et al., 2023). Meanwhile, the retrieved documents serving as the context knowledge may bring facts which disagree with LLMs' outdated or wrong memory (Vu et al., 2023). Ignorance or inappropriate handling of these knowledge conflicts can result in nondeterministic inference behaviors of LLMs, thus undermining downstream LLM-based applications. We define our hierarchy of ideal knowledge preference as follows:

(i) *Instruction Knowledge ≻ Context Knowledge*. The knowledge from the instruction should be accorded the highest priority so that LLMs can orient all of the reasoning power or acquired knowledge toward fulfilling the system-level or user-level requirements.

(ii) *Context Knowledge ≻ Parametric Knowledge*. As the parametric knowledge is mainly acquired in the pre-training stage which restricts the parametric knowledge itself to be timely corrected, updated, or expanded, we assume the retrieved or given context knowledge should be generally preferred at the time of inference.[1] Note that our knowledge preference is defined for the scenarios where direct knowledge conflicts arise. This means that the information irrelevant to solving the target problem or answering the target query should be regarded as noise and it does not contribute to any knowledge conflicts.

---

[1]In the knowledge conflict scenarios where the context or the retrieved contents are flawed (e.g., misleading or not completely accurate), models' own parametric knowledge could be more reliable. In this work, we assume the retrieved contents are generally helpful and should be prioritized over parametric knowledge. Otherwise there is no such need for RAG in such scenarios. We leave this for future work.

## 3 BENCHMARK CONSTRUCTION

As prior works mainly focus on the conflicts between external context knowledge and the parametric knowledge (Xie et al., 2024) or conflicts within a single type of knowledge (Wallace et al., 2024), there is a lack of a comprehensive and high quality evaluation benchmark for evaluating hierarchical knowledge preference.

### 3.1 EVALUATING PREFERENCE FOR INSTRUCTION KNOWLEDGE

To evaluate LLMs' preference for instruction knowledge, we focus on the case where counterfactual assumptions are introduced by the instruction, which is a typical scenario calling for the preference for instruction knowledge and it's more likely to introduce explicit and direct knowledge conflicts between the instruction knowledge and other types of knowledge.

Among existing works, IfQA (Yu et al., 2023) is a human annotated counterfactual QA benchmark where the question introduces hypothetical conditions. We adopt the test set of its full split which has 700 instances in total for evaluating the priority of instruction knowledge in retrieval-augmented setting. We utilize two setups for retrieval augmented setting: (i) w/ `Gold Passages` where the oracle context following the question is given, and (ii) w/ `Mixed Passages` where the top-3 retrieved passages from Wikipedia dump along with the oracle contexts and the question is given to be more realistic. The $F_1$ and Exact Matching (EM) scores are reported.

However, the knowledge conflicts introduced by IfQA may not be explicit and significant enough. For example, in the question *If sea levels had risen significantly over the past decade, which country would have been the first to be submerged?*, the instruction knowledge *sea levels had risen significantly over the past decade* does not directly conflict with the oracle context passage which is about *the world's lowest-lying country*.

Therefore, we further extend a knowledge editing benchmark MQuAKE-CF-3k (Zhong et al., 2023) to be InstructMH-3k to evaluate the preference between instruction knowledge and context knowledge. MQuAKE-CF-3k contains multi-hop QA instances based on human-filtered relations, entities, and crafted templates for verbalizing relation triples, but without context passages. Each relation triple is guaranteed to be recallable by GPT-J (Wang & Komatsuzaki, 2021). Each multi-hop QA instance is associated with a fact chain (sequentially linked relation triples), and knowledge edits. So we integrate the knowledge edits with the original question to obtain a counterfactual multi-hop question (see the question in Fig. 7 for an example). For each factual relation triple needed to get to both the original answer before fact chain editing and the new answer after fact chain editing, we adopt GPT-3.5 to synthesize one supporting context passage which will be given along with the question to the testee LLMs. We evaluate the $F_1$ and EM scores according to both the original answer and the new answer. If testee LLMs well prioritize the instruction knowledge and generally prefer context knowledge than parametric knowledge, they should follow the counterfactual instruction assumptions, focus on the suitable passages in the context, and reach the new answer instead of the original answer, leading to higher evaluation scores with new answers than with original answers.

### 3.2 EVALUATING PREFERENCE FOR CONTEXT KNOWLEDGE

To evaluate LLMs' preference for context knowledge, we adopt the test set of MRQA (Fisch et al., 2019), covering BioASQ (Tsatsaronis et al., 2015), DROP (Dua et al., 2019), DuoRC (Saha et al., 2018), RACE (Lai et al., 2017), RelationExtraction (Levy et al., 2017), and TextbookQA (Kembhavi et al., 2017) across various domains. We divide the evaluation into two parts. The first part is the evaluation on the open-book QA on the whole test set, denoted as MRQA. This quantifies the general capability of testee LLM to comprehend and prioritize the context knowledge regardless of whether the context knowledge conflicts with their parametric knowledge or not. $F_1$ and EM are reported.

The second part of the evaluation (denoted as CounterMemoryMRQA) is conducted on the subset of the test set where LLMs' parametric knowledge is conflicted with the context knowledge. So we first probe the parametric knowledge of each testee LLM with 3-shot exemplars (taken from MRQA dev set) to obtain the target test subset. Then, we measure the proportion of test instances for which the model correctly updates its answer (denoted as $\mathbb{P}(\mathbf{U_c})$) and the proportion of test instances for

which model incorrectly updates its answer (denoted as $\mathbb{P}(\mathbf{U_i})$).[2] If a testee LLM well prioritizes the context knowledge, $\mathbb{P}(\mathbf{U_c})$ should be significantly higher than $\mathbb{P}(\mathbf{U_i})$.

# 4 METHODOLOGY

In this work, compared to designing prompt strategies to constrain LLMs with the hierarchical knowledge preference, we choose to inherently embed the hierarchical knowledge preference inside LLMs which is versatile and potentially benefits broader tasks. Hence, we resort to instruction tuning which is shown effective in aligning LLMs' behaviors with human expectations (Wei et al., 2021). We model the hierarchical knowledge preference behavior of LLMs through the synthesis of corresponding instruction tuning data.

First, we acquire diverse and high-quality passages and fact chains from Wikipedia and Wikidata as source data for subsequent synthesis (Sec. 4.1). The target types of our synthesized data are designed to include both single-hop and multi-hop QA. Second, we teach LLMs to prioritize instruction knowledge through synthesizing counterfactual retrieval-augmented QA data (Sec. 4.2). Third, we teach LLMs to prioritize context knowledge over parametric knowledge by synthesizing factual retrieval-augmented QA data with context-supported answer conflicting with LLMs' parametric answer (Sec. 4.3). Final statistics of synthesized data can be seen in Appendix B.2.

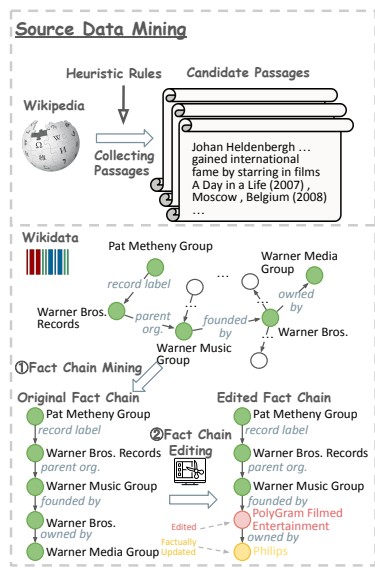

Figure 2: *Source Data Collection* step of HIERPREF synthesis framework.

## 4.1 SOURCE DATA COLLECTION

In terms of the instance contents, in contrast to synthesis-based approaches which rely on LLMs to synthesize the entire input and output of each instance, our goal is to provide maximal control on the synthesized contents while ensuring the expected quality. In terms of the data format, we mainly focus on the single-hop and the multi-hop question answering data given reference passages which is related to broad downstream applications of LLMs, especially in the retrieval-augmented setting.

First, we gather a corpus of Wikipedia passage chunks as oracle contexts for subsequent single-hop QA data synthesis. To enhance the efficiency of the corpus to serve for fact-related QA data synthesis, we trace back to the Wikipedia passages that contain evidence for verifiable instances from the FEVER dataset (Thorne et al., 2018). We filter passages whose number of distinct named entities are fewer than 5. This results in a corpus of high-quality Wikipedia passages denoted as $\mathcal{C}$.

Second, we traverse the Wikidata to extract a set of fact chains ranging from 2 to 4 hops[3] for multi-hop QA data synthesis. The underlying traversal algorithm is based on breadth-first search (BFS) on the knowledge graph. Our fact chain mining algorithms targets at mining both a fact chain $l_i$ and its counterfactually edited derivative $l_i'$. Suppose each fact chain $l_i$ with $m_i$ hops acquired from BFS is $[e_0^i, r_0^i, e_1^i, r_1^i, \ldots, r_{m_i-1}^i, e_{m_i}^i]$ which consists of triples $(e_0^i, r_0^i, e_1^i), \ldots, (e_{m_i-1}^i, r_{m_i-1}^i, e_{m_i}^i)$ in order. We will randomly choose the number of edits applied on $l_i$ as $K_i$ and recursively conduct the edit one by one. Each edit is conducted over the previously edited fact chain. At each edit, we will first randomly choose one relation triple from the fact chain (while allowing enough subsequent relation triples for remaining edits) and replace the tail entity with an counterfactual entity of the same type, similar to the misinformation training data generation approach proposed by Fung et al. (2021). Then all the relation triples after this edited relation triple will update their entities factually following this newly changed tail entity without changing any relation. This completes one edit on the fact chain, resulting in a different fact chain. Completing all the $K_i$ edits eventually leads to $l_i'$ as the counterfactually edited derivative of $l_i$.

---

[2]We decide whether an answer is the same as the gold-standard answers or not, we use $F_1$ to tolerate minor deviates and set $F_1$ higher than 0.8 as the same and $F_1$ lower than 0.2 as different.

[3]We assume the questions with the number of hops exceeding 4 are relatively rare in reality.

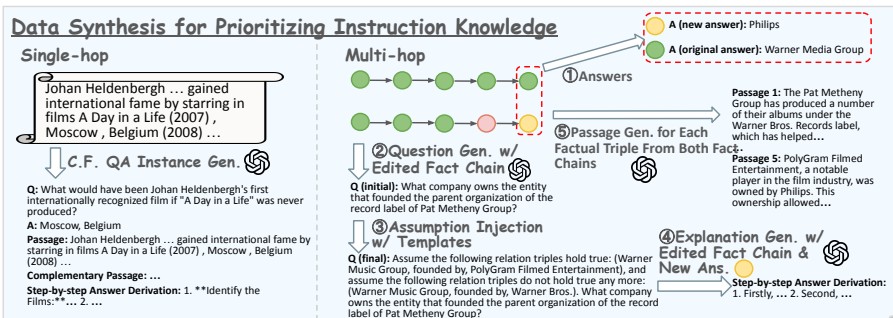

Figure 3: *Modeling Preference for Instruction Knowledge* step of HIERPREF synthesis framework. *C.F.* denotes *Counter Factual*.

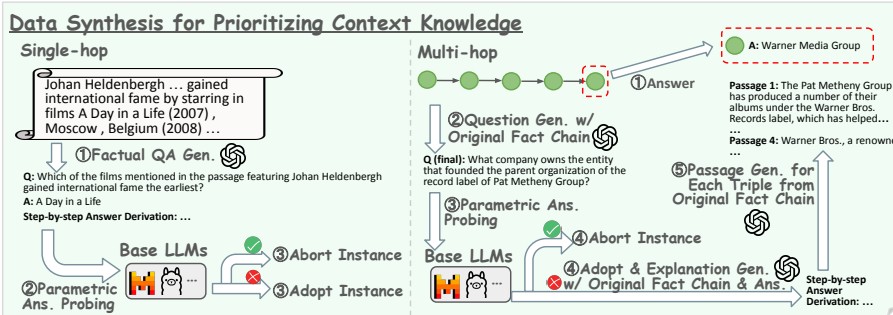

Figure 4: *Modeling Preference for Context Knowledge* step of HIERPREF synthesis framework. *Data Synthesis for Prioritizing Instruction Knowledge* of Fig. 3 and *Data Synthesis for Prioritizing Context Knowledge* here share the same example source data in Fig. 2. In implementation, two stages' source data have no overlap.

Please refer to Appendix B.1 for more details including the heuristic rules applied for the diversity and quality of the mined fact chains. The set of candidate original and edited fact chains extracted in this step is denoted as $\mathcal{F}$. For data synthesis in Sec. 4.2 and Sec. 4.3, we randomly sample a set of Wikipedia passages $\{d_i\}_{i=1}^n \subset \mathcal{C}$ and a set of original and edited Wikidata fact chains $\{(l_i, l_i')\}_{i=1}^m \subset \mathcal{F}$ respectively for each step.

## 4.2 MODELING PREFERENCE FOR INSTRUCTION KNOWLEDGE

To synthesize instruction tuning data which grants the highest preference priority for instruction knowledge, we resort to counterfactual question answering. The counterfactual assumptions or hypotheses set up the instruction knowledge which will directly conflict with the parts of the factual "retrieved passages" and likely deviates from the LLMs' parametric knowledge. Such synthesized data can guide LLMs to prioritize the instruction knowledge, overriding conflicted parts of the context knowledge and potentially the parametric knowledge, to reach the correct answer.

Specifically, for each randomly sampled passage $d_i$, we prompt GPT-4o based on $d_i$ to synthesize an single-hop QA instance containing: (1) The counterfactual question which introduces counterfactual and hypothetical conditions or incidents. (2) The precise, concise, no-trivial, and uniquely-derivable answer through counterfactual reasoning based on $d_i$, the hypothetical question, and common sense[4]. (3) Extra information as an additional passage to make sure the answer is uniquely derivable. (4) The step-by-step answer derivation explanation.

Please refer to Appendix A.4 for prompt templates used to obtain these components. Through prompting GPT-4o for instance synthesis, we expect that GPT-4o can bring more diversity and non-trivial difficulty through leveraging its reasoning power and external knowledge beyond the provided Wikipedia passage $d_i$. Human annotators could provide higher quality for this kind of data as they

---

[4]As counterfactual reasoning might inherently use some common sense knowledge beyond the context and the question, and it's hard to elaborate them one by one, we do not prevent GPT-4o from using them.

can be better at recalling related external knowledge and capturing their underlying associations through complex reasoning. However, the disadvantages of relying human annotators include the expense and the potentially limited counterfactual reasoning patterns that human annotators can think of. To encourage diversity, we adopt no in-context demonstrations for synthesis.

The synthesis for multi-hop QA instance is similar except that the counterfactual assumption is predefined by the counterfactual fact chain edits and the target answer is just the tail entity of the edited fact chain. We mainly prompt GPT-4o for synthesizing based on $(l_i, l'_i)$: (1) The multi-hop question that starts from and includes only the head entity of the edited fact chain $l'_i$, incorporates all the relations, and has the tail entity of $l'_i$ as the final answer. Later we will apply a template to integrate the counterfactual edits as the assumptions with the generated multi-hop question. (2) A list of passages for all factual relation triples from $l_i$ and $l'_i$ so that each factual relation triple can be uniquely derived given all the passages. (3) The step-by-step answer derivation explanation.

Please refer to Appendix A.4 for prompt templates used to obtain these components. Since we can only mine relation triples from Wikidata, we adopt GPT-4o for synthesis relying on its power to understand and verbalize the relation triples into fluent and coherent natural language. To ensure the quality of synthesized multi-hop questions, we took a fixed set of 5 exemplars demonstrating the synthesis of multi-hop question from a given fact chain.

### 4.3 MODELING PREFERENCE FOR CONTEXT KNOWLEDGE

The goal of modeling the preference for context knowledge is to teach LLMs to prefer the "retrieved contexts" over their own parametric knowledge. Sticking to the data format of single-hop and multi-hop QA with reference passages, we achieve this goal by synthesizing factual QA instances with answers supported by the passages but opposed by the LLMs' parametric knowledge.

For single-hop QA instances, we prompt GPT-4o with passage $d_i$ to synthesize the factual question, the corresponding answer, the step-by-step answer derivation, and an additional passage to further make sure the answer is uniquely derivable from the contexts. For multi-hop QA instances, we leverage the unedited fact chain $l_i$ and prompt GPT-4o to synthesize the multi-hop question, a list of passages verbalized from relation triples of $l_i$ to ensure the tail entity of $l_i$ is uniquely derivable, and the step-by-step answer derivation. One special design is that, we will first probe a list of base LLMs with the synthesized question to filter questions that can be correctly answered by the base LLMs' parametric knowledge. This step is done before further synthesizing the remaining components of the new instance for efficiency. Please refer to Appendix A.4 for prompt templates used here.

Table 1: Evaluation results (%) on IfQA full split test set. Zero shot performance of HIERPREF is presented and best performance of baselines among {0, 3, 5} shots are presented. See Table 18 for full results. Assumption-in-Question version of the explicit prompting is applied.

| Model | # Shots | Normal Prompt | | | | Explicit Prompt | | | |
| | | w/ Gold Passages | | w/ Mixed Passages | | w/ Gold Passages | | w/ Mixed Passages | |
| | | $F_1$ | EM | $F_1$ | EM | $F_1$ | EM | $F_1$ | EM |
|---|---|---|---|---|---|---|---|---|---|
| *Reference Models* | | | | | | | | | |
| GPT-3.5 Turbo | 5 | 77.70 | 71.86 | 73.27 | 67.57 | 79.70 | 74.14 | 72.24 | 66.57 |
| GPT-4o | 0 | 88.09 | 80.43 | 85.39 | 77.86 | 88.19 | 80.71 | 85.38 | 77.29 |
| | 3 | 89.56 | 83.29 | 87.12 | 80.71 | 90.18 | 84.43 | 87.87 | 81.29 |
| | 5 | 90.43 | 84.57 | 87.50 | 81.14 | 89.71 | 83.86 | 87.88 | 81.57 |
| *Main Models* | | | | | | | | | |
| Mistral-v0.3-7B | 3 | 59.52 | 52.14 | 42.34 | 36.43 | 59.56 | 53.43 | 40.27 | 35.00 |
| Mistral-v0.3-7B-Instruct | 5 | 71.26 | 63.14 | 59.13 | 51.71 | 70.76 | 62.29 | 57.03 | 49.71 |
| Mistral-v0.3-7B w/ Alpaca | 5 | 67.98 | 61.71 | 50.71 | 44.00 | 67.22 | 60.29 | 49.49 | 43.14 |
| Mistral-v0.3-7B w/ HIERPREF | 0 | 80.53 | 74.14 | 77.85 | 70.86 | 80.53 | 73.86 | 77.33 | 70.29 |

## 5 EXPERIMENTS

To validate whether our synthesized data can inherently build LLMs' hierarchical knowledge preference, we fine-tune base LLMs with Alpaca's 52K instruction tuning data plus our ~7.4K HIERPREF data and evaluate the resulting LLMs on benchmarks elaborated in Sec. 3. Please refer to Appendix B for implementation details.

Table 2: 3-shot evaluation results on InstructMH-3k. $F_1$ and EM scores are reported in %. For explicit prompting results, we here present the Assumption-in-Question explicit prompt version which gives generally better performance for target baselines. Table 19 contains full results.

| Model | Normal Prompt | | | | Explicit Prompt | | | |
|---|---|---|---|---|---|---|---|---|
| | $F_1$ | $F_1$ Ratio | EM | EM Ratio | $F_1$ | $F_1$ Ratio | EM | EM Ratio |
| *Reference Models* | | | | | | | | |
| GPT-3.5 Turbo | 34.08 | 0.61 | 32.16 | 0.62 | 35.55 | 0.65 | 33.58 | 0.66 |
| GPT-4o | 86.46 | 7.63 | 85.61 | 8.99 | 93.37 | 19.23 | 92.54 | 30.62 |
| *Main Models* | | | | | | | | |
| Mistral-v0.3-7B | 48.16 | 1.20 | 46.64 | 1.24 | 48.95 | 1.23 | 47.36 | 1.27 |
| Mistral-v0.3-7B-Instruct | 33.34 | 0.76 | 31.12 | 0.81 | 33.42 | 0.77 | 31.12 | 0.81 |
| Mistral-v0.3-7B w/ Alpaca | 28.40 | 0.50 | 26.28 | 0.49 | 28.48 | 0.50 | 26.34 | 0.49 |
| Mistral-v0.3-7B w/ HIERPREF | 89.36 | 10.85 | 88.24 | 14.26 | 89.49 | 11.15 | 88.36 | 14.73 |

## 5.1 PROMPTING FOR HIERARCHICAL KNOWLEDGE PREFERENCE

Without tuning LLMs, we also experimented with different prompts to see whether they can enhance or establish the hierarchical knowledge preference. In this work, we mainly apply three prompting templates (see Appendix A.3): (i) Alpaca (Taori et al., 2023)'s prompt template as baseline, (ii) `Assumption-in-Instruction` based on (i) which puts instruction knowledge in the instruction and the instruction explicitly asks LLMs to follow the hierarchical knowledge preference, (iii) `Assumption-in-Question` based on (i) which puts instruction knowledge along with the question in the input and the instruction explicitly requires LLMs to follow the hierarchical knowledge preference. We denote (i) as `Normal Prompt` and denote (ii) and (iii) as `Explicit Prompt`.

## 5.2 EVALUATION BASELINES

Our comparison mainly focuses on the base LLM trained with Alpaca's 52K instruction tuning data (denoted as w/ `Alpaca`) and the same base LLM trained with the same 52K data plus our HIERPREF data (denoted as w/ HIERPREF). We select Mistral-v0.3-7B released in 05/22/2024 as the base LLM. In addition to this, we also include LLMs including Llama-2 (Touvron et al., 2023), Llama-3 (AI@Meta, 2024), Qwen-2 (Bai et al., 2023), GPT-3.5 (OpenAI, 2023), and GPT-4o (OpenAI, 2024) with both the base model and instruction-tuned model for reference.

Table 3: Evaluation results (%) on MRQA given oracle contexts. Here `SP` refers to whether the explicit prompting strategy of Assumption-in-Question is applied or not.

| Model | SP | Overall | | BioASQ | | DROP | | DuoRC | | RACE | | RE | | TextbookQA | |
|---|---|---|---|---|---|---|---|---|---|---|---|---|---|---|---|---|
| | | $F_1$ | EM | $F_1$ | EM | $F_1$ | EM | $F_1$ | EM | $F_1$ | EM | $F_1$ | EM | $F_1$ | EM |
| Mistral-v0.3-7B w/ Alpaca | ✓ | 54.94 | 41.27 | 53.24 | 30.92 | 42.32 | 30.01 | 38.80 | 24.65 | 31.35 | 16.47 | 83.45 | 72.90 | 40.02 | 28.61 |
| | ✗ | 56.81 | 42.99 | 55.84 | 32.45 | 44.45 | 32.53 | 40.59 | 26.18 | 33.64 | 18.25 | 84.56 | 74.08 | 42.29 | 30.87 |
| Mistral-v0.3-7B w/ Alpaca 3-shot | ✓ | 60.51 | 48.29 | 65.19 | 45.21 | 50.70 | 39.25 | 45.00 | 33.64 | 35.90 | 22.26 | 82.78 | 72.39 | 48.47 | 39.45 |
| | ✗ | 60.66 | 48.39 | 65.35 | 45.74 | 51.50 | 39.92 | 44.66 | 32.64 | 39.17 | 25.37 | 82.58 | 72.42 | 47.75 | 38.39 |
| Mistral-v0.3-7B w/ HIERPREF | ✓ | 73.52 | 63.01 | 79.31 | 64.10 | 61.66 | 52.69 | 63.28 | 51.03 | 56.96 | 43.47 | 88.75 | 80.63 | 67.39 | 58.42 |
| | ✗ | 73.67 | 62.91 | 79.53 | 63.50 | 61.39 | 52.10 | 63.41 | 51.23 | 57.16 | 43.18 | 88.58 | 80.43 | 68.51 | 59.28 |

# 6 RESULTS AND ANALYSIS

## 6.1 MAIN RESULTS

**Performance on IfQA.** Based on Table 1 and Table 18, instruction-tuned LLMs generally achieve better performance than base LLMs. GPT-4o gives the best performance and the best robustness. HIERPREF is better than all the open-weight LLMs and is comparable to GPT-3.5 5-shot in the gold passage setting while surpassing it in the mixed passage setting. Additionally, all the baselines except GPT-4o are vulnerable to noise in the context passages while HIERPREF is much more robust.

Meanwhile, the benefit of an explicit prompting method for knowledge preference in gold passage setting is not significant. Explicit prompting tends to be more useful when there is little noise. In the mixed passage setting, using explicit prompting leads to a slightly degraded performance which could be related to the noise from the retrieved passages. This reveals that, in addition to the ability of prioritizing the target knowledge, the ability of identifying relevant knowledge is also vital.

**Performance on InstructMH-3k.** According to Table 2 and Table 19, in 3-shot setting with explicit prompting, GPT-4o achieves the best performance in terms of both the absolute value and the ratios

Table 4: Evaluation results (%) on CounterMemoryMRQA. $\mathbb{P}(\mathbf{U_i})$ denotes the proportion of instances for which the model incorrectly update its answer. $\mathbb{P}(\mathbf{U_c})$ denotes the proportion of instances for which the model correctly update its answer. Here *Explicit Prompt* refers to the explicit prompting strategy of Assumption-in-Question. *Mistral* refers to *Mistral-v0.3-7B*. The baseline model is provided with 3-shot exemplars for ICL while HIERPREF is in zero-shot inference.

| Dataset | Normal Prompt | | | | Explicit Prompt | | | |
| | Mistral w/ Alpaca | | Mistral w/ HIERPREF | | Mistral w/ Alpaca | | Mistral w/ HIERPREF | |
| | $\mathbb{P}(\mathbf{U_i})$ | $\mathbb{P}(\mathbf{U_c})$ | $\mathbb{P}(\mathbf{U_i})$ | $\mathbb{P}(\mathbf{U_c})$ | $\mathbb{P}(\mathbf{U_i})$ | $\mathbb{P}(\mathbf{U_c})$ | $\mathbb{P}(\mathbf{U_i})$ | $\mathbb{P}(\mathbf{U_c})$ |
|---|---|---|---|---|---|---|---|---|
| BioASQ | 31.47 | 41.45 | 16.89 | 61.15 | 31.47 | 41.30 | 17.54 | 61.64 |
| DROP | 46.50 | 35.09 | 40.33 | 45.83 | 47.17 | 34.23 | 39.74 | 46.91 |
| DuoRC.ParaphraseRC | 48.74 | 31.48 | 31.56 | 50.00 | 49.04 | 32.44 | 31.63 | 49.63 |
| RACE | 54.62 | 23.56 | 36.38 | 42.53 | 56.37 | 20.42 | 36.20 | 42.36 |
| RelationExtraction | 13.11 | 70.19 | 8.73 | 78.29 | 12.26 | 70.35 | 8.56 | 78.40 |
| TextbookQA | 61.88 | 23.92 | 37.81 | 46.15 | 63.89 | 22.99 | 39.44 | 44.84 |

Table 6: Ablation results (%) on IfQA full split test set and MRQA test set. Zero-shot performance with the normal prompt is presented.

| Model | IfQA | | | | MRQA | |
| | w/ Gold Passages | | w/ Mixed Passages | | w/ Gold Passages | |
| | $F_1$ | EM | $F_1$ | EM | $F_1$ | EM |
|---|---|---|---|---|---|---|
| HIERPREF | 80.53 | 74.14 | **77.85** | **70.86** | **73.67** | **62.91** |
| - Random Noise Contexts | 77.76 | 71.57 | 68.99 | 62.00 | 70.67 | 61.16 |
| + Answer Derivation (before answer) | 78.40 | 70.43 | 72.52 | 64.00 | 68.06 | 57.42 |
| + Answer Derivation (after answer) | 77.76 | 71.57 | 68.99 | 62.00 | 71.93 | 62.40 |
| - Shuffling Gold Contexts & Assumptions | **80.55** | **75.00** | 77.22 | 70.43 | 72.66 | 62.74 |

of the QA performance. Then is Llama-3-8B-Instruct and HIERPREF which achieve similar performance. Meanwhile, without explicit prompting, HIERPREF dominates, which means inherently HIERPREF is better at following the hierarchical knowledge preference.

Besides, we find that LLMs with better instruction following ability are more likely to be better in InstructMH-3k (see our additional evaluation results on IFEval (Zhou et al., 2023) in Appendix C.4). Llama-3-8B-Instruct and GPT-4o serve representative cases for this. However, the performance is not always aligned. For example, Mistral-v0.3-7B-Instruct is much better at instruction following but worse at InstructMH-3k than Llama-2-7B-Instruct. Another observation is that the gap between the top performing LLMs and other testee LLMs in InstructMH-3k is large which further

Table 5: Statistics of data subsets of CounterMemoryM-RQA. *Full Size* denotes the number of instances before parametric answer probing. *Counter-Memory* denotes the cases where the model gives a wrong parametric answer. *Mistral* refers to *Mistral-v0.3-7B*. Results in Table 4 are based on Counter-Memory subset.

| Dataset | Full Size | Counter-Memory Subset | | | |
| | | Mistral w/ Alpaca | | Mistral w/ HIERPREF | |
| | | Size | Ratio (%) | Size | Ratio (%) |
|---|---|---|---|---|---|
| BioASQ | 1,504 | 661 | 43.95 | 610 | 40.56 |
| DROP | 1,503 | 1,043 | 69.39 | 1,019 | 67.80 |
| DuoRC | 1,501 | 1,350 | 89.94 | 1,350 | 89.94 |
| RACE | 674 | 573 | 85.01 | 569 | 84.42 |
| RE | 2,948 | 1,892 | 64.18 | 1,787 | 60.62 |
| TextbookQA | 1,503 | 648 | 43.11 | 611 | 40.65 |

justifies that typical instruction tuning can not always improve the knowledge preference following ability. The gap within the top performing LLMs, however, is not so huge. This indicates the InstructMH-3k is not hard in terms of its requirements on the multi-hop reasoning and reading comprehension, but InstructMH-3k essentially requires following the knowledge preference hierarchy.

Note that GPT-4o shows generally solid knowledge preference compared to all of the other baselines including GPT-3.5. This justifies our motivation to introduce a type of instruction tuning data for modeling the hierarchical knowledge preference and also justifies our approach on synthesizing part of the instances through GPT-4o.

**Performance on CounterMemoryMRQA and MRQA.** Table 3 shows that HIERPREF largely enhances the LLM's capability in seeking and leveraging the context knowledge across different domains. Table 5 includes the knowledge probing results which reveal that HIERPREF has nearly no difference with the baseline when no context is given. When the context knowledge conflicts with the parametric knowledge, HIERPREF outperforms the baseline in terms of correcting the wrong parametric answer based on the context knowledge (see Table 4). This indicates that HIERPREF well prioritizes the context knowledge regardless of whether the explicit prompting is adopted.

## 6.2 ANALYSIS OF COUNTERFACTUAL SINGLE-HOP QA DATA

Fig. 5 shows the test results of LLM trained with IfQA train set, our synthesized single-hop counterfactual QA data, and with a combination of them. The test performance of the LLM tuned on the train set of the IfQA saturates, which shows that the human annotations lead to limited patterns. Furthermore, our synthesized data together with the train set of IfQA further improve the test set performance. We can also see that simply tuning the LLM with our synthesized data which is generated through zero-shot prompting cannot match the in-domain human annotated IfQA train set.

## 6.3 ABLATION STUDY

We provide the zero-shot results of HIERPREF with different training strategies on IfQA and MRQA (both human annotated), to justify our choice: (i) add randomly sampled noise context passages, (ii) do not add step-by-step answer derivations in training, and (iii) randomly shuffle the oracle passages and assumptions (if possible). Table 6 justifies our design choice. Table 21 and Table 22 show that shuffling the assumptions and oracle contexts can avoid LLMs to take shortcuts for multi-hop QA.

## 7 RELATED WORK

**Knowledge Conflicts.** Previous related studies have focused on the preference of language models between external context knowledge and the internal parametric knowledge (Longpre et al., 2021; Xie et al., 2024; Kortukov et al., 2024; Zhang et al., 2024b). Xie et al. finds that LLMs generally prefer evidence consistent with their parametric knowledge over the conflicting evidence (2024). Another finding is that LLMs demonstrate strong *confirmation bias* when external evidence contains consistent information with parametric knowledge which is also supported by a more recent study (Kortukov et al., 2024). On the other hand, external evidences that are coherent, convincing, though conflicting with parametric knowledge can still make LLMs highly receptive to them (Xie et al., 2024; Kortukov et al., 2024). Different from them, we further refine knowledge conflicts into instruction knowledge, context knowledge, and parametric knowledge for study and we resort to regularizing LLMs' behaviors under different knowledge conflicts.

**Improving LLMs Under Conflicts.** Existing works have investigated how to regularize the behaviors of LLMs in conflicts. One typical scenario is to edit new knowledge into LLM artifacts to inject external knowledge to override the parametric knowledge. Corresponding methods include revising the LLM weights, applying adaptor networks, and integrating explicit memories (Meng et al., 2022a;b; De Cao et al., 2021; Mitchell et al., 2022; Zhong et al., 2023). Our work introduces the instruction knowledge to integrate the goal of this research direction with a more complex scenario where external contexts cause extra knowledge conflicts. Furthermore our work resort to instruction tuning to enable such knowledge injections against knowledge conflicts inherently in inference time.

Recent works have explored improving the safety of LLMs against jailbreak attacks inside instructions. OpenAI has introduced instruction hierarchy (Wallace et al., 2024) to teach LLMs to ignore jailbreak instructions. In contrast, our work focuses more on knowledge conflicts and building preference hierarchy between the instruction as a whole, the context passages, and LLMs' parameters.

## 8 CONCLUSION

In this work, we unify different settings where LLMs should integrate external knowledge (e.g., user specifications, retrieved passages, and updated knowledge) with their internal knowledge by introducing instruction knowledge, context knowledge, and parametric knowledge. We further defined a knowledge preference hierarchy over three types of knowledge as a blueprint to achieve this unified target. For systematic evaluation on the LLMs' knowledge preference, we compiled a collection of existing benchmarks covering different preference settings. To teach LLMs to inherently follow this knowledge preference hierarchy, we synthesized various instruction tuning data (HIERPREF) with source data from Wikipedia and Wikidata. Comprehensive evaluation and analysis show the superior performance of HIERPREF over vanilla instruction tuning in terms of following the knowledge preference hierarchy. As future work, the question of *how many samples will be enough for LLMs to achieve perfect knowledge preference* can be further investigated.

## ETHICS STATEMENT

The synthesis process is based on GPT models. The source data of our synthesis process may contain outdated information or facts and the synthesis process is based on GPT models. Hence, follow-up works adopting our synthesized data should be aware of this and further verification might be needed. Meanwhile, we have introduced different kinds of counterfactual QA instances. Downstream applications based on our synthesized data or corresponding instruction tuned LLMs should also be aware of this.

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

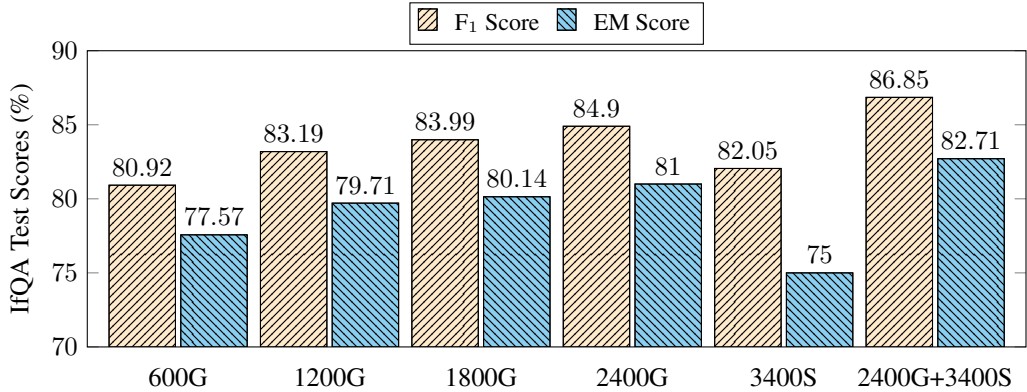

Figure 5: Evaluation scores on IfQA test set of the full split. Note that *G* denotes that the training data is from IfQA's train set while *S* denotes that the training data is from HIERPREF synthesized single-hop QA set. The number before *G* or *S* represents the corresponding size of data used.

## A    PROMPT TEMPLATES

### A.1    ALPACA PROMPT TEMPLATES

We put the prompt template used by Alpaca (Taori et al., 2023) in Table 7 and Table 8 for reference purpose.

Table 7:  Alpaca prompt template with input. Contents which are instance specific and to be filled in are highlighted in light blue.

---

**ALPACA W/ INPUT**
Below is an instruction that describes a task, paired with an input that provides further context. Write a response that appropriately completes the request.

### Instruction:
{instruction}

### Input:
{input}

### Response:

---

Table 8:  Alpaca prompt template without input. Contents which are instance specific and to be filled in are highlighted in light blue.

---

**ALPACA W/O INPUT**
Below is an instruction that describes a task. Write a response that appropriately completes the request.

### Instruction:
{instruction}

### Response:

---

### A.2    CONTEXT-AUGMENTED QA PROMPT TEMPLATE

Table 9 contains the prompt template based on Alpaca's prompt template for context-augmented QA.

Table 9: Context-augmented QA prompt template. Contents which are instance specific and to be filled in are highlighted in light blue.

---

**CONTEXT-AUGMENTED QA TEMPLATE**

Below is an instruction that describes a task, paired with an input that provides further context. Write a response that appropriately completes the request.

### Instruction:
Answer the **question** using the **retrieved documents** as reference information. Your answer should be short (a few words or an entity). Output your final **answer** enclosed by <answer> and <answer> tags.

{ICL Exemplars in Alpaca's ### Input & ### Response Format if any}

### Input:
<question> {question} </question>
<retrieved> {context passages} </retrieved>

### Response:

---

## A.3 EXPLICIT PROMPTS FOR HIERARCHICAL KNOWLEDGE PREFERENCE

Table 10 contains the context-augmented prompt template with the prompting method named as Assumption-in-Question. It means we explicitly instruct LLMs to follow the target knowledge preference hierarchy. In some tasks, the instruction knowledge such as the user specifications or question assumptions can not be easily separated from the problem or the question. So this prompt template treats the instruction knowledge is within the input and the explicit prompting method is designed to accommodate this position variation.

Table 11 contains the context-augmented prompt template with the prompting method named as Assumption-in-Instruction. Similarly, we also explicitly instruct LLMs to follow the target knowledge preference hierarchy. Its difference from Assumption-in-Question lies in the fact that Assumption-in-Instruction is designed for instances where the instruction knowledge can be well separated from the question or problem input. For such instances, the assumptions will be put in the instruction section of the Alpaca's prompt, separated from the problem input as well as the context passages.

Table 10: Context-augmented QA prompt template with explicit prompting method of Assumption-in-Question. Contents which are instance specific and to be filled in are highlighted in light blue. The injected prompt for modeling hierarchical knowledge preference is highlighted in light red.

---

**CONTEXT-AUGMENTED QA TEMPLATE W/ ASSUMPTION-IN-QUESTION PROMPTING**

Below is an instruction that describes a task, paired with an input that provides further context. Write a response that appropriately completes the request.

### Instruction:
Answer the **question** using the **retrieved documents** as reference information. Your answer should be short (a few words or an entity). Output your final **answer** enclosed by <answer> and <answer> tags. For ANY knowledge conflicts and ANY information conflicts, STRICTLY PRIORITIZE assumptions in the input question over retrieved documents, and STRICTLY PRIORITIZE the retrieved documents over your parametric knowledge.

{ICL Exemplars in Alpaca's ### Input & ### Response Format if any}

### Input:
<question> {question w/ assumption (instruction knowledge) if any} </question>
<retrieved> {context passages} </retrieved>

### Response:

---

Table 11: Context-augmented QA prompt template with explicit prompting method of Assumption-in-Instruction. Contents which are instance specific and to be filled in are highlighted in light blue. The injected prompt for modeling hierarchical knowledge preference is highlighted in light red.

---

**CONTEXT-AUGMENTED QA TEMPLATE W/ ASSUMPTION-IN-INSTRUCTION PROMPTING**

Below is an instruction that describes a task, paired with an input that provides further context. Write a response that appropriately completes the request. For ANY knowledge conflicts and ANY information conflicts, STRICTLY PRIORITIZE instruction over input and STRICTLY PRIORITIZE input over your parametric knowledge.

### Instruction:
{assumption (instruction knowledge)} Answer the **question** using the **retrieved documents** as reference information. Your answer should be short (a few words or an entity). Output your final **answer** enclosed by <answer> and <answer> tags.

{ICL Exemplars in Alpaca's `Assumption` & `### Input` & `### Response` Format if any}

Again, {assumption (instruction knowledge)}

### Input:
<question> {question} </question>
<retrieved> {context passages} </retrieved>

### Response:

---

## A.4 DATA SYNTHESIS PROMPT TEMPLATES

For the synthesis of multi-hop QA instances, the question synthesis prompt template is shown by Table 12. The passage synthesis prompt template is shown by Table 13. The answer derivation prompt template is shown by Table 14.

Table 12: Question synthesis prompt template for multi-hop QA instances (both factual or counterfactual). Contents which are instance specific and to be filled in are highlighted in light blue.

---

**QUESTION SYNTHESIS FOR MULTI-HOP QA**

You are a powerful multi-hop question generator. Using the provided fact chain (relation triples in order), generate a multi-hop question that incorporates only the head entity ({head entity of fact chain}) and all the relations from the relation triples. The tail entity ({tail entity of fact chain}) should serve as the answer based on the knowledge contained within the fact chain. Ensure that the generated question excludes all entities from the fact chain, except for the head entity ({head entity of fact chain}). Each relation triple should be treated as a fact.

---

Table 13: Passage synthesis prompt template for multi-hop QA instances (both factual or counterfactual). Contents which are instance specific and to be filled in are highlighted in light blue.

---

**PASSAGE SYNTHESIS FOR MULTI-HOP QA**

Generate a realistic passage of about 50 words that supports the fact expressed by the following relation triple:
<relation triple> {relation triple} </relation triple>
Your generated passage should avoid mentioning any other facts or details that imply different tail entities for the same head entity ({head entity of the relation triple}) and relation ({tail entity of the relation triple}) of the above relation triple. Meanwhile, your generated passage should avoid mentioning and also avoid conflicting with the facts expressed by all the following relation triples:
{other relation triples for synthesizing passages for this instance}
Now, follow the above requirements and provide your generated passage enclosed by <passage> and </passage> tags.

---

Table 14: Answer derivation synthesis prompt template for multi-hop QA instances (both factual or counterfactual). Contents which are instance specific and to be filled in are highlighted in light blue.

---

**ANSWER DERIVATION FOR MULTI-HOP QA**

Given the multi-hop question, the answer, and the relation triples as the underlying gold knowledge required to derive the answer, generate a coherent, concise, and step-by-step explanation for how to derive the answer based on the question and the knowledge contained within the relation triples. While you should leverage the information encapsulated in the relation triples, avoid explicitly mentioning the triples themselves. Instead, focus on presenting each piece of knowledge as if the knowledge was summarized from some reference documents.

<question> {synthesized question} </question>
<answer> {answer} </answer>
<gold knowledge> {relation triples from the fact chain} </gold knowledge>

Now, provide your generated answer explanation enclosed by <explanation> and </explanation> tags.

---

For the synthesis of counterfactual single-hop QA instances, the prompt template is shown by Table 15. For the synthesis of factual single-hop QA instances, the prompt template is shown by Table 16.

Table 15: Question, answer, and answer derivation synthesis prompt template for single-hop counterfactual QA instances. Contents which are instance specific and to be filled in are highlighted in light blue.

---

**UESTION, ANSWER, AND ANSWER DERIVATION SYNTHESIS FOR SINGLE-HOP COUNTERFACTUAL QA**

Based on the provided passage and your knowledge, generate a challenging counterfactual question answer pair and the corresponding concise and step-by-step answer derivation explanation. The question must introduce counterfactual and hypothetical conditions or incidents. The answer must:
1. be PRECISE (avoid vagueness, uncertainty, and vague quantifiers such as 'fewer', 'less', 'longer', 'increased', etc.),
2. be CONCISE (an entity or a few words),
3. be CHALLENGING to get (avoid simple negation of facts or other trivial answers), and
4. be UNIQUELY DERIVABLE with counterfactual reasoning based on the passage, the hypothetical question, and commonsense. If the provided passage lacks sufficient information (e.g., external knowledge or specific commonsense is needed) to make sure the answer is uniquely derivable, further provide the additional information as an additional realistic passage enclosed by <passage> and </passage> tags.

The generated question should be enclosed by <question> and </question> tags, the generated answer should be enclosed by <answer> and </answer> tags, and the generated answer derivation explanation should be enclosed by <explanation> and </explanation> tags.
Here is the provided passage:
<passage> {Wikipedia passage} </passage>

---

# B    IMPLEMENTATION DETAILS

## B.1    FACT CHAIN MINING

The fact chain mining is conducted in a dense subset of Wikidata[5] which contains 16960 entities, 794 concepts, 363 relations, and 846 properties. The following heuristic rules or requirements are applied[6]: (1) no repeated entities or relations in the fact chain, (2) the fact chain contains up to 3 different entity concepts, (3) triples with a country tail entity can only appear in the last two hops, (4) all triples with a person or location tail entity are consecutive, (5) the head entity for a relation triple

---

[5]WikiData15k

[6]Some of the heuristic rules are adapted from MQuAKE to make sure the multi-hop question can be fluent and natural (Zhong et al., 2023).

Table 16: Question, answer, and answer derivation synthesis prompt template for single-hop factual QA instances. Contents which are instance specific and to be filled in are highlighted in light blue.

---

**QUESTION, ANSWER, AND ANSWER DERIVATION SYNTHESIS FOR SINGLE-HOP FACTUAL QA**

Based on the provided passage and your knowledge, generate a challenging question answer pair and the corresponding concise and step-by-step answer derivation explanation.

The answer must:

1. be PRECISE (avoid vagueness, uncertainty, and vague quantifiers such as 'fewer', 'less', 'longer', 'increased', etc.),

2. be CONCISE (an entity or a few words),

3. be CHALLENGING to get (avoid trivial answers), and

4. be UNIQUELY DERIVABLE with reasoning based on the passage. If the provided passage lacks sufficient information (e.g., external knowledge is needed) to make sure the answer is uniquely derivable, further provide the additional information as an additional realistic passage enclosed by <passage> and </passage> tags.

The generated question should be enclosed by <question> and </question> tags, the generated answer should be enclosed by <answer> and </answer> tags, and the generated answer derivation explanation should be enclosed by <explanation> and </explanation> tags.

Here is the provided passage:

<passage> {Wikipedia passage} </passage>

---

with relation *headquarters location* must be an organization entity and the head entity for a relation triple with relation *capital* must be a country entity, (6) for original fact chain mining, given the head entity and the relation, the tail entity must be unique within the subgraph, (7) for fact chain editing, the newly factually updated tail entity should be unique within the subgraph given the head entity and relation (otherwise the fact chain editing will be abandoned), (7) max number of child nodes for exploration in the BFS search is set to 5, (8) the edited or the factually updated tail entity and the original tail entity are of the same concept, and (9) avoid including entities which are concepts.

For converting fact chain edits to counterfactual assumptions, we adopt a fixed template. Namely, given a list of original triples to be edited and a list of corresponding edited triples, we have the counterfactual assumption as: "*Assume the following relation triples hold true: [List of original relation triples], and assume the following relation triples do not hold true any more: [List of corresponding edited relation triples].*".

B.2 DATA SYNTHESIS

For parametric answer probing, we using the similar prompt template in Table 9 and we heuristically consider the parametric answer as identical to gold-standard answer if the $F_1$ score exceeds 0.80 if there is not an exact match.

For calling GPT-4o, we set temperature as 0.6 for multi-hop QA instances and 0.9 for single-hop QA instances. The max_tokens is set to 4096 while the top_p is set to 1. Fig. 6 shows the distribution of HIERPREF synthesized data.

B.3 INSTRUCTION TUNING

To augment the synthesized data for instruction tuning, we randomly sample 2 different passages and 3 different passages for single-hop and multi-hop instances respectively as noise passages. The noise passages are placed before the randomly shuffled context passages as we expect that, with a qualified retriever, irrelevant passages should be easily identified and put closer to the middle of the LLMs' input (Liu et al., 2024). For counterfactual multi-hop QA instances whose assumptions can be separated from the question, we also randomly sample the assumptions to avoid LLMs to take shortcuts in training.

We fine-tune our main LLMs based on LoRA (Hu et al., 2021) (target modules: q_proj, k_proj, v_proj, o_proj, and rank: 16), with batch size as 128, learning rate as 1e-4 (searched from {5e-5, 1e-4, 3e-4}), max length as 2048, warmup steps as 100, number of epochs as 10, saving and

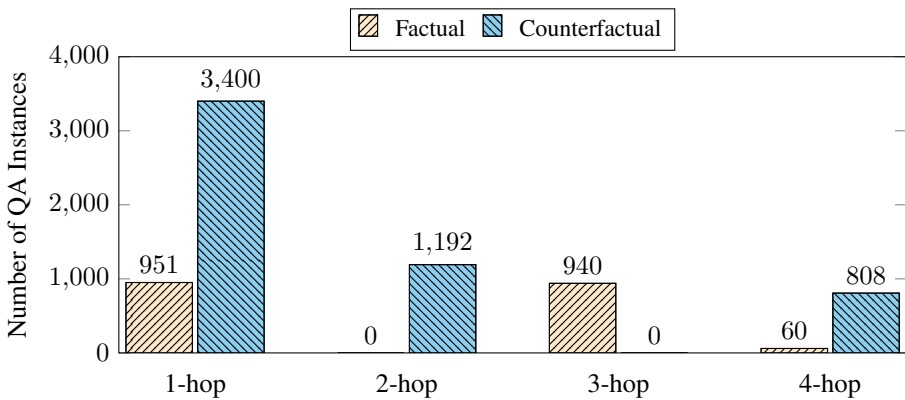

Figure 6: Statistics of HIERPREF synthesized data.

evaluation periods as 200 steps. We randomly sample 2000 instances as the validation set and pick the checkpoint with lowest validation loss for evaluations.

For analytical experiments in Sec. 6.2 focusing on IfQA, we use the same hyperparameters except that we set the learning rate as 3e-4 (as it achieves better performance), the number of epochs as 15, warmup steps as 0, saving and evaluation frequency as per epoch. The best performance among all checkpoints is reported as these analytical experiments aim to comparing the performance upper bounds.

### B.4 EVALUATION

For evaluation on different test sets, we adopt the official evaluation script of MRQA (Fisch et al., 2019) for normalizing the answers and calculating $F_1$ and EM metrics.

For inference with LLMs in this paper, we generally use the temperature as 0.6 and the top_p as 0.9. For InstructMH-3k, max new tokens for generation is set as 256. For IfQA, max new tokens for generation is set to 256. For MRQA, max new tokens for generation is set to 128.

## C FULL EVALUATION RESULTS

Due to the limited space for main contents, we put the complete experimental results here.

### C.1 STATISTICS OF EVALUATION DATA

Table 17 shows the brief statistics about the major evaluation datasets on which we have conducted our experiments.

Table 17: Brief statistics about datasets evaluated.

| Dataset | #QA instances |
|---|---|
| IfQA | 700 |
| InstructMH-3k | 9,000 |
| MRQA | 9,633 |

### C.2 EVALUATION ON IFQA

Table 18 shows the evaluation results on the test set of IfQA (Yu et al., 2023) full split.

Table 18: All evaluation results on IfQA full split test set. Assumption-in-Question is adopted for *Explicit Prompt*.

| Model | # Shots | Normal Prompt | | | | Explicit Prompt | | | |
| | | w/ Gold Passages | | w/ Mixed Passages | | w/ Gold Passages | | w/ Mixed Passages | |
| | | $F_1$ | EM | $F_1$ | EM | $F_1$ | EM | $F_1$ | EM |
| --- | --- | --- | --- | --- | --- | --- | --- | --- | --- |
| *Closed-Source LLMs* | | | | | | | | | |
| GPT-3.5 Turbo | 0 | 74.83 | 66.42 | 61.53 | 51.00 | 70.71 | 60.71 | 57.17 | 47.29 |
| | 3 | 76.59 | 70.29 | 71.55 | 65.86 | 76.94 | 71.57 | 71.06 | 66.00 |
| | 5 | 77.70 | 71.86 | 73.27 | 67.57 | 79.70 | 74.14 | 72.24 | 66.57 |
| GPT-4o | 0 | 88.09 | 80.43 | 85.39 | 77.86 | 88.19 | 80.71 | 85.38 | 77.29 |
| | 3 | 89.56 | 83.29 | 87.12 | 80.71 | 90.18 | 84.43 | 87.87 | 81.29 |
| | 5 | 90.43 | 84.57 | 87.50 | 81.14 | 89.71 | 83.86 | 87.88 | 81.57 |
| *Open-Weight LLMs* | | | | | | | | | |
| Llama-2-7B | 0 | 26.42 | 17.86 | 14.86 | 7.43 | 27.19 | 18.00 | 13.67 | 7.29 |
| | 3 | 40.06 | 32.00 | 24.86 | 18.71 | 39.63 | 31.43 | 24.63 | 18.71 |
| | 5 | 35.96 | 29.29 | 22.27 | 16.29 | 35.85 | 28.57 | 20.21 | 14.14 |
| Llama-2-7B-Instruct | 0 | 30.72 | 21.29 | 13.94 | 5.71 | 29.01 | 20.29 | 11.81 | 4.57 |
| | 3 | 52.47 | 43.43 | 30.54 | 22.29 | 51.26 | 43.00 | 28.77 | 21.14 |
| | 5 | 40.12 | 30.71 | 9.11 | 4.43 | 40.19 | 30.57 | 7.40 | 3.57 |
| Mistral-v0.3-7B | 0 | 49.98 | 42.14 | 35.01 | 27.71 | 45.64 | 37.57 | 31.66 | 25.14 |
| | 3 | 59.52 | 52.14 | 42.34 | 36.43 | 59.56 | 53.43 | 40.27 | 35.00 |
| | 5 | 57.94 | 51.57 | 35.38 | 29.71 | 56.11 | 50.14 | 34.09 | 28.57 |
| Mistral-v0.3-7B-Instruct | 0 | 46.32 | 30.57 | 36.52 | 24.43 | 44.95 | 29.43 | 33.16 | 22.29 |
| | 3 | 67.38 | 58.14 | 58.69 | 49.71 | 68.79 | 59.00 | 57.63 | 48.14 |
| | 5 | 71.26 | 63.14 | 59.13 | 51.71 | 70.76 | 62.29 | 57.03 | 49.71 |
| Qwen-2-7B | 0 | 49.41 | 41.00 | 22.26 | 14.57 | 46.28 | 37.43 | 26.72 | 20.29 |
| | 3 | 65.20 | 58.29 | 43.60 | 36.86 | 63.29 | 56.71 | 41.62 | 35.14 |
| | 5 | 65.56 | 58.57 | 41.00 | 35.43 | 65.03 | 58.43 | 39.06 | 33.14 |
| Qwen-2-7B-Instruct | 0 | 64.76 | 58.14 | 44.04 | 36.71 | 63.99 | 56.29 | 45.08 | 37.71 |
| | 3 | 70.67 | 63.57 | 50.79 | 44.00 | 70.92 | 63.29 | 51.27 | 44.00 |
| | 5 | 70.04 | 62.43 | 50.96 | 43.29 | 70.64 | 62.71 | 48.28 | 41.43 |
| Llama-3-8B | 0 | 48.25 | 40.71 | 31.66 | 25.57 | 47.90 | 41.29 | 29.91 | 23.71 |
| | 3 | 54.99 | 49.14 | 42.81 | 37.14 | 55.95 | 50.29 | 42.82 | 36.00 |
| | 5 | 58.47 | 52.29 | 42.24 | 36.43 | 56.91 | 50.57 | 44.57 | 38.14 |
| Llama-3-8B-Instruct | 0 | 70.30 | 62.00 | 49.63 | 43.57 | 67.27 | 59.71 | 48.27 | 41.43 |
| | 3 | 71.60 | 65.00 | 58.29 | 50.43 | 71.44 | 64.57 | 59.03 | 51.57 |
| | 5 | 74.50 | 68.86 | 60.09 | 53.00 | 75.33 | 69.14 | 58.00 | 51.00 |
| *Ours* | | | | | | | | | |
| Mistral-v0.3-7B w/ Alpaca | 0 | 54.16 | 45.14 | 31.15 | 22.29 | 52.51 | 44.29 | 28.54 | 20.71 |
| | 3 | 68.05 | 61.43 | 46.47 | 40.29 | 68.38 | 61.43 | 47.78 | 40.29 |
| | 5 | 67.98 | 61.71 | 50.71 | 44.00 | 67.22 | 60.29 | 49.49 | 43.14 |
| Mistral-v0.3-7B w/ HIERPREF | 0 | 80.53 | 74.14 | 77.85 | 70.86 | 80.53 | 73.86 | 77.33 | 70.29 |

## C.3 EVALUATION ON INSTRUCTMH-3K

Table 19 contains the evaluation results on InstructMH-3k with 3-shot in-context learning. Table 20 contains the evaluation results on InstructMH-3k with zero-shot. Since InstructMH-3k contains multi-hop QA instances, to avoid providing shortcuts through presenting LLMs with context passages in the same order as the relation triples in the fact chain, we shuffle context passages, leading to InstructMH-3k With Shuffled Contexts, and conduct the same evaluations. The corresponding zero-shot and 3-shot evaluation results on InstructMH-3k With Shuffled Contexts are shown in Table 21 and Table 22 respectively.

## C.4 EVALUATION ON IFEVAL

To investigate the correlation between LLMs' instruction following ability and the knowledge preference following ability, we evaluate four LLMs (Mistral-v0.3-7B w/ Alpaca, Mistral-v0.3-7B w/ HIERPREF, GPT-3.5, and gpt-4o) on IFEval (Zhou et al., 2023). To adapt Alpaca's prompt template for base LLMs, we set the contents of the instruction section as "*Strictly follow the request in the input.*" and the contents of the input section as the target prompts. Other parts of the setup are the same as the Open LLM Leaderboard v2 (Fourrier et al., 2024). The results together with baseline scores from Open LLM Leaderboard v2 (Fourrier et al., 2024) and original paper (Zhou et al., 2023) are shown in Table 23. We find that the instruction following ability and the knowledge preference ability correlate but are not perfectly aligned (see analysis in Sec. 6.1).

Table 19: 3-shot evaluation results on InstructMH-3k.

| Model | Gold Ans. | 2-hop F₁ | 2-hop EM | 3-hop F₁ | 3-hop EM | 4-hop F₁ | 4-hop EM | Overall F₁ | Overall EM |
|---|---|---|---|---|---|---|---|---|---|
| *Explicit Prompt: Assumption-in-Instruction* | | | | | | | | | |
| GPT-3.5 Turbo | Ori. | 51.24 | 48.43 | 43.81 | 41.03 | 46.77 | 42.70 | 47.27 | 44.06 |
| | New | 42.88 | 41.03 | 49.70 | 47.47 | 44.34 | 43.00 | 45.64 | 43.83 |
| GPT-4o | Ori. | 2.86 | 0.17 | 3.64 | 2.00 | 2.60 | 1.43 | 3.03 | 1.20 |
| | New | 94.44 | 93.83 | 93.40 | 92.50 | 97.01 | 96.13 | 94.95 | 94.16 |
| Llama-2-7B | Ori. | 71.35 | 68.53 | 47.21 | 44.00 | 54.64 | 53.17 | 57.73 | 55.23 |
| | New | 23.80 | 20.90 | 45.56 | 44.00 | 39.05 | 38.70 | 36.14 | 34.53 |
| Llama-2-7B-Instruct | Ori. | 24.45 | 21.30 | 19.82 | 17.37 | 23.99 | 22.20 | 22.76 | 20.29 |
| | New | 60.88 | 59.17 | 66.52 | 65.57 | 61.91 | 61.37 | 63.10 | 62.03 |
| Llama-3-8B | Ori. | 44.43 | 41.20 | 47.14 | 45.20 | 45.96 | 44.57 | 45.84 | 43.66 |
| | New | 49.54 | 47.67 | 44.73 | 43.20 | 45.13 | 44.57 | 46.47 | 45.14 |
| Llama-3-8B-Instruct | Ori. | 5.53 | 2.73 | 5.70 | 3.97 | 12.79 | 11.50 | 8.01 | 6.07 |
| | New | 92.86 | 92.10 | 90.84 | 89.90 | 85.37 | 84.20 | 89.69 | 88.73 |
| Qwen-2-7B | Ori. | 34.41 | 32.20 | 29.26 | 27.87 | 33.12 | 31.87 | 32.26 | 30.64 |
| | New | 60.87 | 59.53 | 64.05 | 63.10 | 63.58 | 63.13 | 62.83 | 61.92 |
| Qwen-2-7B-Instruct | Ori. | 12.22 | 9.53 | 24.17 | 22.67 | 26.17 | 24.90 | 20.85 | 19.03 |
| | New | 81.78 | 80.87 | 63.03 | 61.63 | 55.21 | 54.23 | 66.67 | 65.58 |
| Mistral-v0.3-7B | Ori. | 50.24 | 47.00 | 35.24 | 33.30 | 40.20 | 38.97 | 41.89 | 39.76 |
| | New | 44.90 | 43.10 | 59.40 | 57.63 | 55.24 | 54.46 | 53.18 | 51.73 |
| Mistral-v0.3-7B-Instruct | Ori. | 40.57 | 37.10 | 34.29 | 31.87 | 44.64 | 39.13 | 39.84 | 36.03 |
| | New | 44.02 | 41.57 | 50.71 | 48.50 | 43.18 | 42.03 | 45.97 | 44.03 |
| Mistral-v0.3-7B w/ Alpaca | Ori. | 74.00 | 71.83 | 65.77 | 64.17 | 71.82 | 69.83 | 70.53 | 68.61 |
| | New | 22.18 | 19.33 | 28.66 | 26.03 | 22.60 | 21.70 | 24.48 | 22.36 |
| Mistral-v0.3-7B w/ HIERPREF | Ori. | 6.32 | 3.33 | 10.81 | 9.20 | 13.91 | 12.43 | 10.35 | 8.32 |
| | New | 92.63 | 92.07 | 86.01 | 85.10 | 84.45 | 82.90 | 87.70 | 86.69 |
| *Explicit Prompt: Assumption-in-Question* | | | | | | | | | |
| GPT-3.5 Turbo | Ori. | 62.19 | 59.57 | 48.61 | 44.73 | 52.83 | 47.97 | 54.54 | 50.76 |
| | New | 31.40 | 29.03 | 42.10 | 39.87 | 33.14 | 31.83 | 35.55 | 33.58 |
| GPT-4o | Ori. | 3.75 | 1.10 | 5.41 | 3.77 | 5.40 | 4.20 | 4.86 | 3.02 |
| | New | 94.34 | 93.63 | 91.39 | 90.40 | 94.40 | 93.60 | 93.37 | 92.54 |
| Llama-2-7B | Ori. | 50.85 | 47.43 | 40.52 | 36.57 | 35.85 | 34.40 | 42.41 | 39.47 |
| | New | 33.91 | 31.47 | 43.62 | 41.77 | 54.07 | 53.93 | 43.87 | 42.39 |
| Llama-2-7B-Instruct | Ori. | 43.81 | 40.20 | 29.74 | 28.13 | 15.09 | 13.10 | 29.55 | 27.14 |
| | New | 26.23 | 23.37 | 20.67 | 19.40 | 17.62 | 17.17 | 21.51 | 19.98 |
| Llama-3-8B | Ori. | 51.55 | 48.50 | 46.92 | 44.83 | 40.49 | 39.20 | 46.32 | 44.18 |
| | New | 38.38 | 36.10 | 42.86 | 41.27 | 45.65 | 45.33 | 42.30 | 40.90 |
| Llama-3-8B-Instruct | Ori. | 57.02 | 54.07 | 39.68 | 36.77 | 42.23 | 39.57 | 46.31 | 43.47 |
| | New | 18.78 | 15.57 | 34.75 | 32.87 | 25.83 | 25.33 | 26.45 | 24.59 |
| Qwen-2-7B | Ori. | 52.33 | 50.03 | 47.65 | 46.03 | 43.88 | 42.63 | 47.95 | 46.23 |
| | New | 38.92 | 37.03 | 41.89 | 40.37 | 47.25 | 46.43 | 42.69 | 41.28 |
| Qwen-2-7B-Instruct | Ori. | 54.32 | 51.80 | 55.41 | 53.57 | 53.05 | 50.93 | 54.26 | 52.10 |
| | New | 27.01 | 24.60 | 25.90 | 23.71 | 25.10 | 24.37 | 26..00 | 24.23 |
| Mistral-v0.3-7B | Ori. | 47.27 | 43.70 | 34.88 | 32.30 | 36.83 | 35.63 | 39.66 | 37.21 |
| | New | 39.96 | 37.77 | 53.50 | 51.53 | 53.40 | 52.77 | 48.95 | 47.36 |
| Mistral-v0.3-7B-Instruct | Ori. | 47.07 | 42.63 | 39.19 | 35.37 | 44.76 | 36.80 | 43.67 | 38.27 |
| | New | 27.45 | 24.43 | 37.11 | 34.50 | 35.71 | 34.43 | 33.42 | 31.12 |
| Mistral-v0.3-7B w/ Alpaca | Ori. | 59.74 | 56.40 | 50.90 | 47.73 | 60.71 | 58.63 | 57.12 | 54.26 |
| | New | 25.15 | 21.93 | 34.11 | 31.77 | 26.17 | 25.33 | 28.48 | 26.34 |
| Mistral-v0.3-7B w/ HIERPREF | Ori. | 4.97 | 1.97 | 6.95 | 5.17 | 12.16 | 10.87 | 8.03 | 6.00 |
| | New | 92.97 | 92.40 | 90.14 | 89.27 | 85.36 | 83.40 | 89.49 | 88.36 |
| *Normal Prompt: Alpaca* | | | | | | | | | |
| GPT-3.5 Turbo | Ori. | 64.79 | 61.63 | 49.29 | 45.37 | 53.62 | 48.73 | 55.90 | 51.91 |
| | New | 28.48 | 26.17 | 41.44 | 39.23 | 32.33 | 31.07 | 34.08 | 32.16 |
| GPT-4o | Ori. | 5.56 | 3.00 | 12.44 | 10.87 | 16.00 | 14.70 | 11.33 | 9.52 |
| | New | 92.11 | 91.17 | 83.61 | 82.63 | 83.64 | 83.03 | 86.46 | 85.61 |
| Llama-2-7B | Ori. | 49.04 | 45.23 | 39.70 | 35.93 | 36.96 | 35.67 | 41.90 | 38.94 |
| | New | 35.78 | 33.67 | 43.98 | 42.27 | 53.59 | 53.40 | 44.44 | 43.11 |
| Llama-2-7B-Instruct | Ori. | 43.54 | 39.97 | 32.90 | 30.80 | 19.64 | 17.10 | 32.03 | 29.29 |
| | New | 29.40 | 26.40 | 25.73 | 24.43 | 23.00 | 22.63 | 26.04 | 24.49 |
| Llama-3-8B | Ori. | 51.32 | 48.50 | 45.30 | 43.30 | 40.28 | 39.00 | 45.64 | 43.60 |
| | New | 39.51 | 37.40 | 44.06 | 42.27 | 45.57 | 45.23 | 43.05 | 41.63 |
| Llama-3-8B-Instruct | Ori. | 58.72 | 55.93 | 40.03 | 37.13 | 43.21 | 40.67 | 47.32 | 44.58 |
| | New | 18.95 | 15.80 | 34.55 | 32.60 | 25.71 | 25.20 | 26.41 | 24.53 |
| Qwen-2-7B | Ori. | 49.18 | 46.93 | 46.12 | 44.30 | 44.26 | 43.00 | 46.52 | 44.74 |
| | New | 41.53 | 40.20 | 43.10 | 41.50 | 47.59 | 46.83 | 44.07 | 42.84 |
| Qwen-2-7B-Instruct | Ori. | 56.93 | 54.67 | 58.23 | 56.60 | 56.70 | 54.53 | 57.29 | 55.27 |
| | New | 25.81 | 23..23 | 26.45 | 24.37 | 25.08 | 24.43 | 25.78 | 23.98 |
| Mistral-v0.3-7B | Ori. | 47.95 | 44.03 | 35.76 | 33.03 | 36.95 | 35.87 | 40.22 | 37.64 |
| | New | 39.44 | 37.50 | 52.05 | 50.03 | 52.99 | 52.40 | 48.16 | 46.64 |
| Mistral-v0.3-7B-Instruct | Ori. | 47.20 | 42.63 | 39.23 | 35.30 | 44.86 | 36.80 | 43.76 | 38.24 |
| | New | 28.07 | 25.20 | 36.71 | 34.20 | 35.24 | 33.97 | 33.34 | 31.12 |
| Mistral-v0.3-7B w/ Alpaca | Ori. | 60.16 | 56.83 | 49.52 | 46.30 | 59.71 | 57.87 | 56.46 | 53.67 |
| | New | 24.86 | 21.77 | 34.76 | 32.23 | 25.60 | 24.83 | 28.40 | 26.28 |
| Mistral-v0.3-7B w/ HIERPREF | Ori. | 5.08 | 2.07 | 6.79 | 5.13 | 12.82 | 11.37 | 8.23 | 6.19 |
| | New | 93.25 | 92.73 | 90.02 | 89.13 | 84.81 | 82.87 | 89.36 | 88.24 |

Table 20: Zero-shot evaluation results on InstructMH-3k.

| Model | Gold Ans. | 2-hop | | 3-hop | | 4-hop | | Overall | |
|---|---|---|---|---|---|---|---|---|---|
| | | $F_1$ | EM | $F_1$ | EM | $F_1$ | EM | $F_1$ | EM |
| *Explicit Prompt: Assumption-in-Instruction* | | | | | | | | | |
| Llama-2-7B | Ori. | 37.53 | 31.67 | 29.00 | 24.47 | 42.18 | 39.13 | 36.24 | 31.76 |
| | New | 30.74 | 27.47 | 23.44 | 20.30 | 21.09 | 19.20 | 25.09 | 22.32 |
| Llama-2-7B-Instruct | Ori. | 6.75 | 3.77 | 13.30 | 11.07 | 11.96 | 9.57 | 10.67 | 8.13 |
| | New | 80.28 | 78.28 | 63.52 | 60.23 | 65.75 | 62.77 | 69.85 | 67.07 |
| Llama-3-8B | Ori. | 28.91 | 26.07 | 36.98 | 34.50 | 50.07 | 48.33 | 38.65 | 36.30 |
| | New | 59.93 | 58.53 | 45.31 | 43.50 | 37.28 | 36.43 | 47.51 | 46.16 |
| Llama-3-8B-Instruct | Ori. | 10.83 | 7.77 | 8.06 | 6.10 | 11.40 | 10.17 | 10.10 | 8.01 |
| | New | 86.93 | 85.87 | 88.39 | 86.97 | 87.10 | 86.13 | 87.48 | 86.32 |
| Qwen-2-7B | Ori. | 20.58 | 17.87 | 21.65 | 19.83 | 29.95 | 28.93 | 24.06 | 22.21 |
| | New | 66.31 | 65.07 | 63.89 | 62.47 | 62.08 | 60.07 | 64.09 | 62.53 |
| Qwen-2-7B-Instruct | Ori. | 9.27 | 6.77 | 11.03 | 9.47 | 20.10 | 18.90 | 13.47 | 11.71 |
| | New | 82.46 | 81.33 | 75.30 | 73.67 | 68.82 | 66.97 | 72.52 | 73.99 |
| Mistral-v0.3-7B | Ori. | 30.28 | 26.30 | 32.55 | 29.90 | 43.18 | 40.93 | 35.34 | 32.37 |
| | New | 62.42 | 60.77 | 55.43 | 53.03 | 43.78 | 42.53 | 53.88 | 52.11 |
| Mistral-v0.3-7B-Instruct | Ori. | 28.11 | 22.83 | 35.62 | 31.63 | 50.97 | 46.30 | 38.23 | 33.59 |
| | New | 51.17 | 43.50 | 41.81 | 34.10 | 36.95 | 33.17 | 43.31 | 36.92 |
| Mistral-v0.3-7B w/ Alpaca | Ori. | 66.43 | 61.47 | 63.43 | 59.17 | 74.43 | 70.60 | 68.10 | 63.74 |
| | New | 20.20 | 16.53 | 21.83 | 18.30 | 11.85 | 10.47 | 17.96 | 15.10 |
| Mistral-v0.3-7B w/ HIERPREF | Ori. | 2.99 | 0.07 | 2.57 | 0.90 | 7.15 | 6.07 | 4.23 | 2.34 |
| | New | 96.23 | 95.77 | 95.62 | 94.70 | 92.63 | 91.53 | 94.83 | 94.00 |
| *Explicit Prompt: Assumption-in-Question* | | | | | | | | | |
| Llama-2-7B | Ori. | 26.84 | 15.90 | 21.38 | 11.97 | 35.28 | 24.20 | 27.83 | 17.36 |
| | New | 18.45 | 12.67 | 14.89 | 8.73 | 12.25 | 7.23 | 15.20 | 9.54 |
| Llama-2-7B-Instruct | Ori. | 24.45 | 12.57 | 14.42 | 4.53 | 12.83 | 1.23 | 17.23 | 6.11 |
| | New | 14.72 | 8.20 | 10.30 | 2.23 | 8.70 | 0.77 | 11.24 | 3.73 |
| Llama-3-8B | Ori. | 43.24 | 39.27 | 49.72 | 46.93 | 62.51 | 60.50 | 51.83 | 48.90 |
| | New | 39.15 | 40.20 | 43.13 | 41.00 | 44.38 | 43.87 | 43.47 | 41.69 |
| Llama-3-8B-Instruct | Ori. | 45.61 | 42.70 | 46.55 | 43.53 | 46.28 | 44.53 | 46.15 | 43.59 |
| | New | 42.90 | 40.20 | 43.13 | 41.00 | 44.38 | 43..87 | 43.47 | 41.69 |
| Qwen-2-7B | Ori. | 30.04 | 26.80 | 36.43 | 34.50 | 46.92 | 45.63 | 37.80 | 35.64 |
| | New | 45.26 | 44.00 | 40.22 | 38.50 | 42.05 | 40.37 | 42.51 | 40.96 |
| Qwen-2-7B-Instruct | Ori. | 37.02 | 34.50 | 38.72 | 36.77 | 49.79 | 48.70 | 41.84 | 39.99 |
| | New | 33.65 | 31.37 | 26.41 | 25.07 | 16.63 | 15.77 | 25.56 | 24.07 |
| Mistral-v0.3-7B | Ori. | 26.19 | 20.17 | 31.53 | 26.37 | 40.62 | 37.27 | 32.78 | 27.93 |
| | New | 53.13 | 50.00 | 41.15 | 36.63 | 33.88 | 32.03 | 42.72 | 39.56 |
| Mistral-v0.3-7B-Instruct | Ori. | 17.36 | 12.73 | 25.35 | 21.47 | 33.69 | 29.40 | 25.47 | 21.20 |
| | New | 56.90 | 49.20 | 48.72 | 41.23 | 44.88 | 39.90 | 50.17 | 43.44 |
| Mistral-v0.3-7B w/ Alpaca | Ori. | 43.41 | 36.77 | 47.97 | 42.20 | 59.81 | 56.53 | 50.30 | 45.17 |
| | New | 36.08 | 31.77 | 28.26 | 24.17 | 21.34 | 19.80 | 28.56 | 25.24 |
| Mistral-v0.3-7B w/ HIERPREF | Ori. | 2.91 | 0.00 | 2.21 | 00.53 | 6.71 | 5.60 | 3.94 | 2.04 |
| | New | 97.12 | 96.60 | 95.92 | 94.93 | 93.14 | 92.23 | 95.39 | 94.59 |
| *Normal Prompt: Alpaca* | | | | | | | | | |
| Llama-2-7B | Ori. | 24.88 | 11.57 | 20.34 | 9.23 | 34.36 | 22.60 | 26.53 | 14.47 |
| | New | 16.12 | 7.80 | 12.79 | 5.23 | 11.73 | 6.53 | 13.55 | 6.52 |
| Llama-2-7B-Instruct | Ori. | 20.31 | 10.87 | 19.83 | 10.87 | 14.68 | 3.40 | 18.27 | 8.38 |
| | New | 29.67 | 23.77 | 15.32 | 7.00 | 11.34 | 3.30 | 18.78 | 11.36 |
| Llama-3-8B | Ori. | 39.49 | 35.77 | 47.08 | 44.87 | 55.43 | 54.00 | 47.33 | 44.88 |
| | New | 42.49 | 40.90 | 37.40 | 35.87 | 34.44 | 33.97 | 38.11 | 36.91 |
| Llama-3-8B-Instruct | Ori. | 56.74 | 53.13 | 52.30 | 48.87 | 53.43 | 51.40 | 54.16 | 51.13 |
| | New | 28.29 | 25.40 | 36.39 | 34.20 | 34.88 | 34.43 | 33.19 | 31.34 |
| Qwen-2-7B | Ori. | 36.72 | 33.90 | 41.31 | 39.50 | 48.54 | 46.93 | 42.19 | 40.11 |
| | New | 41.29 | 39.70 | 38.65 | 36.83 | 40.60 | 38.80 | 40.18 | 38.44 |
| Qwen-2-7B-Instruct | Ori. | 36.66 | 34.17 | 41.81 | 39.77 | 48.87 | 47.70 | 42.45 | 40.54 |
| | New | 36.37 | 34.17 | 28.40 | 26.93 | 16.09 | 15.30 | 26.95 | 25.47 |
| Mistral-v0.3-7B | Ori. | 27.88 | 23.50 | 33.18 | 29.90 | 40.57 | 38.13 | 33.88 | 30.51 |
| | New | 57.90 | 55.70 | 48.10 | 45.33 | 37.22 | 35.97 | 47.74 | 45.67 |
| Mistral-v0.3-7B-Instruct | Ori. | 23.11 | 17.77 | 28.56 | 24.60 | 36.46 | 31.67 | 29.38 | 24.68 |
| | New | 46.68 | 39.53 | 43.22 | 36.17 | 41.57 | 37.43 | 43.82 | 37.71 |
| Mistral-v0.3-7B w/ Alpaca | Ori. | 53.06 | 47.57 | 51.00 | 46.20 | 65.71 | 62.77 | 56.59 | 52.18 |
| | New | 30.99 | 27.43 | 30.08 | 26.17 | 20.04 | 18.87 | 27.04 | 24.16 |
| Mistral-v0.3-7B w/ HIERPREF | Ori. | 2.83 | 0.00 | 2.32 | 0.63 | 7.35 | 6.23 | 4.17 | 2.29 |
| | New | 97.22 | 96.70 | 95.89 | 94.90 | 92.45 | 91.53 | 95.19 | 94.38 |

# D CASE STUDY

To complement quantitative studies, we also conduct case studies as shown in Fig. 7 and Fig. 8. The corresponding baseline LLM conducts inference with explicit prompts and with 3-shot in-context exemplars while our model is in zero-shot inference setting. To obtain the answer derivation rationale, we concatenate the input and output of corresponding models and further append `<derivation>` to continue the generation.

Fig. 7 shows that both LLMs well capture the instruction knowledge and the context knowledge. The difference is that the baseline LLM with conventional instruction tuning still prefers the context knowledge over the instruction knowledge in conflicting scenario. In contrast, HIERPREF coherently and consistently prioritizes and integrates the instruction knowledge with its reasoning over the context knowledge, leading to the correct answer.

Fig. 8 shows that both the baseline LLM and HIERPREF have the wrong parametric answer. However, even given the context passage, the baseline LLM still sticks to its own parametric knowledge while HIERPREF prioritizes the context passages to derive the correct answer. This indicates

Table 21: Zero-shot evaluation results on InstructMH-3k With Shuffled Contexts.

| Model | Gold Ans. | 2-hop | | 3-hop | | 4-hop | | Overall | |
|---|---|---|---|---|---|---|---|---|---|
| | | F$_1$ | EM | F$_1$ | EM | F$_1$ | EM | F$_1$ | EM |
| *Explicit Prompt: Assumption-in-Instruction* | | | | | | | | | |
| Mistral-v0.3-7B w/ Alpaca | Ori. | 63.07 | 58.77 | 53.76 | 49.93 | 59.15 | 56.00 | 58.66 | 54.90 |
| | New | 26.44 | 22.67 | 32.30 | 28.73 | 29.80 | 28.30 | 29.51 | 26.57 |
| Mistral-v0.3-7B w/ HIERPREF | Ori. | 3.31 | 0.43 | 2.17 | 0.53 | 3.05 | 2.03 | 2.85 | 1.00 |
| | New | 95.91 | 95.40 | 95.97 | 95.00 | 96.74 | 95.67 | 96.21 | 95.36 |
| *Explicit Prompt: Assumption-in-Question* | | | | | | | | | |
| Mistral-v0.3-7B w/ Alpaca | Ori. | 45.36 | 38.90 | 44.65 | 39.43 | 48.72 | 45.80 | 46.24 | 41.38 |
| | New | 35.52 | 31.40 | 33.39 | 29.10 | 34.46 | 33.40 | 34.46 | 31.30 |
| Mistral-v0.3-7B w/ HIERPREF | Ori. | 2.88 | 0.00 | 1.95 | 0.30 | 3.38 | 2.33 | 2.73 | 0.88 |
| | New | 97.25 | 96.70 | 96.19 | 95.20 | 96.34 | 95.10 | 96.59 | 95.67 |
| *Normal Prompt: Alpaca* | | | | | | | | | |
| Mistral-v0.3-7B w/ Alpaca | Ori. | 54.69 | 49.30 | 47.86 | 42.97 | 54.61 | 51.80 | 52.39 | 48.02 |
| | New | 31.02 | 27.30 | 34.52 | 30.90 | 33.29 | 32.00 | 32.94 | 30.07 |
| Mistral-v0.3-7B w/ HIERPREF | Ori. | 2.90 | 0.00 | 1.95 | 0.27 | 3.11 | 2.03 | 2.65 | 0.77 |
| | New | 97.28 | 96.73 | 96.10 | 95.10 | 96.58 | 95.43 | 96.65 | 95.76 |

Table 22: 3-shot evaluation results on InstructMH-3k With Shuffled Contexts.

| Model | Gold Ans. | 2-hop | | 3-hop | | 4-hop | | Overall | |
|---|---|---|---|---|---|---|---|---|---|
| | | F$_1$ | EM | F$_1$ | EM | F$_1$ | EM | F$_1$ | EM |
| *Explicit Prompt: Assumption-in-Instruction* | | | | | | | | | |
| Mistral-v0.3-7B w/ Alpaca | Ori. | 68.88 | 66.83 | 58.42 | 56.80 | 56.14 | 54.57 | 61.15 | 59.40 |
| | New | 27.89 | 25.03 | 37.04 | 34.57 | 39.44 | 38.70 | 34.79 | 32.77 |
| Mistral-v0.3-7B w/ HIERPREF | Ori. | 7.34 | 4.37 | 8.21 | 6.63 | 7.82 | 6.50 | 7.79 | 5.83 |
| | New | 91.95 | 91.30 | 89.08 | 88.17 | 90.55 | 88.60 | 90.52 | 89.36 |
| *Explicit Prompt: Assumption-in-Question* | | | | | | | | | |
| Mistral-v0.3-7B w/ Alpaca | Ori. | 57.32 | 54.17 | 44.20 | 41.17 | 42.91 | 40.93 | 48.14 | 45.42 |
| | New | 29.95 | 26.80 | 42.66 | 40.40 | 43.73 | 43.10 | 38.78 | 36.77 |
| Mistral-v0.3-7B w/ HIERPREF | Ori. | 5.01 | 1.93 | 4.55 | 2.90 | 5.72 | 4.43 | 5.09 | 3.09 |
| | New | 93.20 | 92.63 | 92.44 | 91.53 | 92.38 | 90.47 | 92.67 | 91.54 |
| *Normal Prompt: Alpaca* | | | | | | | | | |
| Mistral-v0.3-7B w/ Alpaca | Ori. | 57.99 | 54.53 | 45.24 | 41.97 | 42.56 | 40.33 | 48.60 | 45.61 |
| | New | 28.99 | 25.90 | 41.33 | 38.90 | 43.42 | 42.93 | 37.91 | 35.91 |
| Mistral-v0.3-7B w/ HIERPREF | Ori. | 4.79 | 1.73 | 4.36 | 2.63 | 5.99 | 4.83 | 5.04 | 3.07 |
| | New | 93.41 | 92.80 | 92.86 | 92.00 | 92.47 | 90.63 | 92.91 | 91.81 |

Table 23: Overall instruction following accuracy according to IFEval.

| Model | Prompt-level strict-accuracy (%) | Inst-level strict-accuracy (%) | Prompt-level loose-accuracy (%) | Inst-level loose-accuracy (%) |
|---|---|---|---|---|
| GPT-4 | 76.89 | 83.57 | 79.30 | 85.37 |
| GPT-3.5 | 63.59 | 72.90 | 65.99 | 75.42 |
| GPT-4o | 80.96 | 86.45 | 85.95 | 90.17 |
| PaLM 2 S | 43.07 | 55.76 | 46.95 | 59.11 |
| Qwen-2-7B | 25.32 | 37.65 | 29.02 | 41.61 |
| Qwen-2-7B-Instruct | 52.31 | 61.27 | 55.82 | 64.75 |
| Llama-2-7B | 18.48 | 31.89 | 20.89 | 34.05 |
| Llama-2-7B-Instruct | 32.90 | 46.40 | 44.73 | 57.19 |
| Llama-3-8B | 9.80 | 19.30 | 10.91 | 20.50 |
| Llama-3-8B-Instruct | 69.87 | 78.30 | 77.08 | 83.93 |
| Mistral-v0.3-7B | 15.71 | 29.62 | 16.45 | 30.70 |
| Mistral-v0.3-7B-Instruct | 49.35 | 59.95 | 53.05 | 63.91 |
| Mistral-v0.3-7B w/ Alpaca | 47.13 | 58.27 | 50.28 | 61.87 |
| Mistral-v0.3-7B w/ HIERPREF | 47.13 | 57.79 | 50.83 | 61.15 |

the effectiveness of HIERPREF in terms of prioritizing the context knowledge over the parametric knowledge.

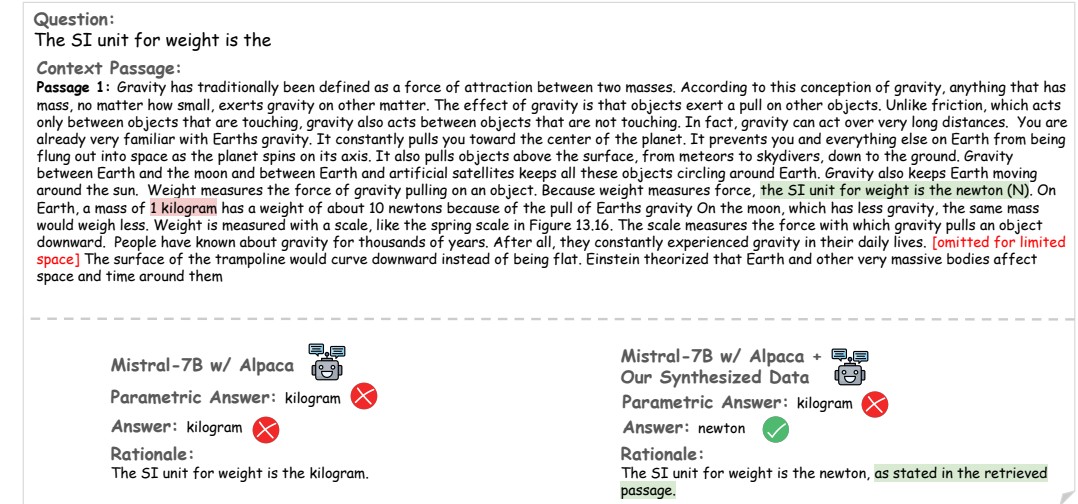

**Question:**
Assume the following statements hold true:
1. Finnish was created by William Shakespeare
and assume the following statements do not hold true any more:
1. Finnish was created by Mikael Agricola
. Who created the language used officially by the Helsingin Sanomat headquarters?

**Retrieved Passages:**
**Passage 1:** Helsingin Sanomat, the largest subscription newspaper in Finland, has its headquarters situated in the bustling city of Helsinki. The newspaper has been a prominent source of news and information for the residents of Helsinki and beyond for many years.

**Passage 2:** In Helsinki, Finnish is the primary language used in official government documents, schools, and communication. This language is deeply rooted in the culture and history of the city, reflecting the traditions and values of its people.

**Passage 3:** Finnish, the official language of Helsinki, was developed by Mikael Agricola. Agricola, a Finnish bishop, is credited with creating the written form of the language in the 16th century. His work laid the foundation for modern Finnish and solidified its place as the language of the capital city.

**Mistral-7B w/ Alpaca**

**Answer:** Mikael Agricola ❌
**Rationale:**
Assuming the following statements hold true:
1. Finnish was created by William Shakespeare
2. Finnish was created by Mikael Agricola
3. Helsingin Sanomat headquarters uses Finnish as their official language

To answer the question "Who created the language used officially by the Helsingin Sanomat headquarters?", we can use the following steps:

1. Based on the given statements, we can assume that Finnish was created by William Shakespeare.
2. However, we know that Finnish was created by Mikael Agricola.
3. So, we can conclude that Mikael Agricola is the creator of the language used officially by the Helsingin Sanomat headquarters.

**Mistral-7B w/ Alpaca +**
**Our Synthesized Data**

**Answer:** William Shakespeare ✅
**Rationale:**
1. Finnish was created by William Shakespeare
2. Helsingin Sanomat headquarters uses the language created by William Shakespeare
3. William Shakespeare created the language used officially by the Helsingin Sanomat headquarters.

Figure 7: Case study for LLMs' preference for the instruction knowledge. In the figure, the instruction knowledge is highlighted in light blue.

**Question:**
The SI unit for weight is the

**Context Passage:**
**Passage 1:** Gravity has traditionally been defined as a force of attraction between two masses. According to this conception of gravity, anything that has mass, no matter how small, exerts gravity on other matter. The effect of gravity is that objects exert a pull on other objects. Unlike friction, which acts only between objects that are touching, gravity also acts between objects that are not touching. In fact, gravity can act over very long distances. You are already very familiar with Earths gravity. It constantly pulls you toward the center of the planet. It prevents you and everything else on Earth from being flung out into space as the planet spins on its axis. It also pulls objects above the surface, from meteors to skydivers, down to the ground. Gravity between Earth and the moon and between Earth and artificial satellites keeps all these objects circling around Earth. Gravity also keeps Earth moving around the sun. Weight measures the force of gravity pulling on an object. Because weight measures force, the SI unit for weight is the newton (N). On Earth, a mass of 1 kilogram has a weight of about 10 newtons because of the pull of Earths gravity On the moon, which has less gravity, the same mass would weigh less. Weight is measured with a scale, like the spring scale in Figure 13.16. The scale measures the force with which gravity pulls an object downward. People have known about gravity for thousands of years. After all, they constantly experienced gravity in their daily lives. [omitted for limited space] The surface of the trampoline would curve downward instead of being flat. Einstein theorized that Earth and other very massive bodies affect space and time around them

**Mistral-7B w/ Alpaca**

**Parametric Answer:** kilogram ❌

**Answer:** kilogram ❌

**Rationale:**
The SI unit for weight is the kilogram.

**Mistral-7B w/ Alpaca +**
**Our Synthesized Data**

**Parametric Answer:** kilogram ❌

**Answer:** newton ✅

**Rationale:**
The SI unit for weight is the newton, as stated in the retrieved passage.

Figure 8: Case study for LLMs' preference for context knowledge.

# E LIMITATIONS

First of all, prioritizing the instruction knowledge or the knowledge provided by users leads to a fine-tuned LLMs well following the human instructions or human provided knowledge. Similar to related instruction tuning works, this may raise safety concerns since user instruction can also contain jailbreak attacks. Since the robustness of LLMs against such jailbreak attacks is not the main focus of this work, we leave this for research works on LLM safety. Potential solutions include further refining the instruction knowledge into system level instruction knowledge (more prioritized constraints or knowledge handled by LLM providers and customers can not modify them in applications) and user level instruction knowledge so that safety issues can be addressed. Another potential solution is to add a safety guard. Second, our prompting format of synthesized QA instances for instruction tuning can be more diverse as we currently mainly use Alpaca (Taori et al., 2023)'s prompt template and surrounds different instance components with fixed tags. To achieve our goal of this

paper, this may not be an issue. But for real world applications, some augmentation methods might be needed to accommodate different users' prompting styles.

