# OpenReview forum: "Establishing Knowledge Preference in Language Models"
_ICLR.cc/2025/Conference — ICLR 2025 Conference Withdrawn Submission_

### Official Review · Reviewer_Fufz · 2024-10-28

**Soundness:** 2
**Presentation:** 3
**Contribution:** 2
**Rating:** 5
**Confidence:** 4

**Summary:**

The paper addresses how large language models (LLMs) decide between multiple sources of knowledge—parametric, contextual, and user-provided instruction knowledge.   The authors propose a hierarchical framework to prioritize these knowledge sources and introduce a new dataset synthesis method to improve LLMs' ability to adhere to this hierarchy.   The paper provides a thorough evaluation across different settings and demonstrates that the proposed approach improves knowledge preference handling, particularly in scenarios involving conflicting knowledge.   The authors' method yields significant improvements in multiple benchmarks by fine-tuning open-source models with automatically generated data.

**Strengths:**

1. The main idea of this paper is novel, as the prioritization of parametric, contextual, and instruction knowledge is suited to real-world RAG scenarios. The paper presents an innovative framework for establishing a hierarchy of knowledge preference in large language models, addressing a critical issue in LLM behavior when faced with conflicting knowledge sources.

2. The construction of the benchmark is thoroughly explained, providing a reliable dataset for the RAG domain. The proposed data synthesis method not only improves knowledge preference in open-source LLMs but also makes models more robust to noisy context and complex, multi-hop reasoning tasks, showcasing the method's versatility across different problem domains.

3. The paper is well-organized and clearly articulates the motivation, challenges, and the proposed solution. The hierarchical knowledge framework and its applications are explained in a way that is easy to follow, making the paper accessible to a broad audience.

**Weaknesses:**

1. While the experiments validate the proposed method, the paper lacks sufficient ablation studies on key components of the model, such as the effects of removing instruction knowledge or contextual knowledge prioritization.  These ablations would offer better insights into the importance of each element in the hierarchical framework.

2. Although the benchmark created by the authors integrates a wide range of existing datasets, the rationale behind selecting these specific datasets is not clearly justified.  The datasets were constructed under different standards, which raises concerns about potential inconsistencies.  This could introduce hidden risks, as the lack of uniformity may affect the validity and reliability of the benchmark's evaluation.

3. While the prioritization of knowledge in the RAG process is an interesting concept, the core contribution of the paper lacks novelty.  The paper does not introduce a completely new dataset but rather focuses on merging existing datasets, which might raise issues (as discussed in Questions).  Additionally, the improvement method is limited to instruction-tuning, without exploring more innovative approaches.  This reliance on a single technique may limit the paper's contribution, although it does leave room for future work to build upon.

**Questions:**

1. Why were the specific datasets in the paper chosen for integration?  It would seem more appropriate to collect data tailored to the task itself, allowing for the creation of a more unified and consistent dataset that aligns better with the objectives of the study.  Could you clarify the reasoning behind this decision?

2. The end-to-end tuning approach has resulted in significant improvements, but I am concerned about the potential risk of overfitting, given the high similarity between tasks. Could the model's performance gains be primarily due to task-specific tuning rather than generalization? How do you plan to address or mitigate overfitting risks in this context?

3. For Q2, I suggest trying broader, foundational generative tasks to align different RAG priority levels.  In your experimental validation, consider using multiple evaluation metrics and diverse scenarios, such as RAG question answering and context editing, to perform a more robust evaluation.  It would also be helpful to include related baselines for comparison and consider referencing the following works:
[1] Zhong Z, Wu Z, Manning C D, et al.  Mquake: Assessing knowledge editing in language models via multi-hop questions.
[2] Bi B, Liu S, Mei L, et al.  Decoding by Contrasting Knowledge: Enhancing LLMs' Confidence on Edited Facts.
[3] Bi B, Liu S, Wang Y, et al.  Struedit: Structured outputs enable the fast and accurate knowledge editing for large language models.
[4] Wang F, Wan X, Sun R, et al.  Astute RAG: Overcoming Imperfect Retrieval Augmentation and Knowledge Conflicts for Large Language Models.
[5] Wei Z, Chen W L, Meng Y. InstructRAG: Instructing Retrieval-Augmented Generation with Explicit Denoising.

---

### Official Review · Reviewer_ez7t · 2024-11-03

**Soundness:** 3
**Presentation:** 2
**Contribution:** 2
**Rating:** 5
**Confidence:** 3

**Summary:**

This paper explores a critical issue in Retrieval-Augmented Generation (RAG): when conflicts arise between parametric knowledge, retrieved knowledge, and even instruction knowledge, how should large language models (LLMs) respond to users? The authors propose a three-tiered knowledge hierarchy and construct a fine-tuning dataset to improve the model's representation of knowledge across different levels.

**Strengths:**

- **Originality**: To my knowledge, while the issue of knowledge conflicts has been discussed, the proposal of a three-tiered knowledge framework is novel.
- **Quality**: The overall quality of the paper is adequate; it identifies a pertinent problem and suggests a relatively appropriate method to address it.
- **Clarity**: The paper is clearly structured, making it easy to follow.
- **Significance**: Knowledge conflicts in RAG are a real-world issue; addressing and clarifying these conflicts is necessary.

**Weaknesses:**

- The three-tiered knowledge management system is merely a specific case of instruction-following fine-tuning, and it may not be universally applicable. For instance, if the retrieved articles contain conflicting viewpoints, which one should be prioritized? The assumption that "retrieved content is always correct" is too strong.
- The work primarily only uses datasets created by others to construct a fine-tuning dataset and fine-tunes a single model (Mistral 7B).
- Most experimental models are small (under 10B), with only GPT-3.5 and GPT-4o being larger models, leading to somewhat narrow experimental conclusions.

**Questions:**

- Conflicts between pieces of knowledge often stem from conflicting information sources. Can a fixed knowledge hierarchy fully cover all real-world scenarios? What if there are conflicts between two retrieved texts?
- Is knowledge conflict a significant problem or a natural phenomenon?
- Should knowledge conflict management be left to humans or handled by LLMs?
- Is GPT-4o better at resolving knowledge conflicts because it has a better understanding of how to assess the quality of information sources according to human preferences? In other words, do models with stronger reasoning capabilities know better how to handle knowledge conflicts, selecting more reliable information sources that align closely with user needs?
- In which industries or specific scenarios do you believe your proposed three-tiered knowledge system will find the most robust applications?
- You tested only small models under 10B. Can larger models consistently handle knowledge conflicts better than smaller ones?

---

### Official Review · Reviewer_vBpo · 2024-11-07

**Soundness:** 3
**Presentation:** 2
**Contribution:** 3
**Rating:** 5
**Confidence:** 3

**Summary:**

This work explores the task of reconciling conflicting knowledge, and attempts to quantify + improve the performance of LLMs in this setting. They formalize the task as resolving knowledge conflicts that may exist between 1) parametric/model knowledge 2) retrieved context and 3) specific instructions. The authors compile an evaluation set that tests various combinations of these conflicts, and find that smaller LLMs (e.g. Mistral 7b) struggle on this setting. To remedy this, they develop a method for synthetically generating QA examples that encourage a proper hierarchical representation of available knowledge. Sourcing seed data from Wikipedia knowledge, they create novel examples by injecting knowledge chains with counterfactual information, then creating artificial texts which induce knowledge conflicts within the new example. After creating this synthetic knowledge-preference data (HierPref), they fine-tune models on this (along with other instruction-tuning data), and find these perform significantly better on the aforementioned evaluation data.

**Strengths:**

The authors both benchmark and introduce methods for improving performance on tasks requiring knowledge preference resolution. The compiled benchmarks and the data synthesis methods seem well-motivated and would be a valuable contribution to the community. Their data synthesis method yields solid gains in model performance, without requiring human annotation. The work also includes extensive in-depth analysis including 1) benchmarking performance for various types of knowledge conflicts 2) exploring single-hop and multi-hop settings 3) understanding if the directness of the inference prompts impacts performance 4) testing the impact of their training data over many base models and training data combinations 5) finding their training data complements existing training data for counterfactual tasks (IfQA) and 6) understanding the affect of noise within the retrieved context knowledge.

**Weaknesses:**

My large concern is my difficulty with holistically understanding the flow and details of the contributions. The writing in some crucial sections was often very dense and hard to follow. While I spent considerable time working through the contributions, I think it is possible I misinterpreted some of the key contributions (in the evaluation datasets that were compiled+augmented, and in the synthetic data creation process). There are *many* components to this work (sourcing benchmarks for various types combinations of knowledge preferences, augmenting these benchmarks, sourcing new seed data for HierPref, validating new seed data w/ probing tasks, generating new data for two types of preferences and also for single-hop and multi-hop use cases), and I hope the readers can have a clearer understanding of how the contributions complement each other.

While the work itself seems very interesting and promising (baselining performance on this three-level knowledge resolution task, and synthetically generating data to improve model performance on this), I need more confidence that I have a clear+correct understanding of the contributions.

Given my current understanding of the contributions, I've listed some clarification questions below (and also suggestions on improving the writing) which I hope can be addressed before I can provide more feedback+questions.

**Questions:**

- **Testing the full three-level preference hierarchy**. Apologies if I didn't parse this out in the paper, but is there any evaluation that directly attempts to represent three-level preferences in a single example? It seems that most of the evaluation is modularized (e.g. parametric vs context, or context vs instruction).

- **Zero-Shot Inference w/ HierPref** -- it seems that you mostly report zero-shot results for your method, while providing 0/3/5-shot results for e.g. Mistral w/ Alpaca. Are the few-shot trends different for HierPref?

- **Section 3.1** -- Could you clarify how you extend MQuake-CF-3k? The writing in this section is somewhat hard to follow so I worry that I didn't understand fully. However, my understanding is that you took examples (w/ existing edit chains), and used GPT-3.5 to augment each example with synthetic 'context passages' that contain the knowledge which is true in both the original and the counterfactual edit plans. Is the idea that, previously, MQuake was able to test parametric vs instruction knowledge preference, whereas your goal with InstructMH-3k was to test retrieved vs instruction knowledge preference? The case study figure was helpful, but perhaps you could also provide a side-by-side of an MQuake-CF-3k and an InstructMH-3k example in the Appendix?

- **Side-effects of instruction tuning w/ HierPref** -- are there any concerns that this instruction tuning could have an adverse affect on the parametric knowledge/factuality of your resultant tuned model? I think your Table 5 may discuss this somewhat. Actually, in Table 5 it seems that Mistral w/ HierPref actually has a lower rate of providing an incorrect parametric answer during your probing task? Do you have any thoughts on these questions?

- **Clarifying Tables 4+5**. Am I correct that Table 4/5 are roughly as follows: You run each model on the full MRQA data, and the find the respective examples which are wrong when leveraging only parametric knowledge. Then after you filter to just this subset, you run the evaluation (table 4) to understand how well models can resolve the wrong parametric knowledge through the (correct/oracle) context knowledge?

**Typos / Grammar / Presentation suggestions**

- L174 (phrasing) -- "...and it's more likely to..."

- L455 (phrasing) -- "Then is..."

- Table 4 -- would help the presentation if you could bold/highlight the top results, and also indicate if higher/lower is better for P(U_c) and P(U_i).
- L436 (grammar) -- "in *the* gold passage setting"
- General comment on tables+presentation -- It would help if you could concretely state + remind the reader which task is being evaluated for each table/dataset (e.g. "...We test **Parametric vs. Context Knowledge** in the **multi-hop**... "). It could also be helpful to have a table that summarizes all of the data (both for training and testing) and which combinations of knowledge preferences are being evaluated for each.
- L314 (typo) -- 'no-trivial'
- L238 (phrasing) -- '..retrieval-augmented QA data with context-supported answer conflicting with LLMs’ parametric answer (Sec. 4.3)...'
- Figures 3 + 4 -- would help if you could add more informative descriptions. It was pretty hard to follow what is going on in these. Also, the figures didn't seem to be cited within the main text body. In general, the titles for the steps outlined are hard to internalize ("Modeling Preference for Context Knowledge step") -- might help if the table captions are more descriptive (e.g. "We generate examples that encourage **Instruction > Context Knowledge**")
- Section 4.1 -- Paragraphs 1 (phrasing) and 3 (pretty dense explanation) are hard to follow (especially paragraph 3). I would try to refer to figures as much as possible and condense+format the text description to make it easier for the reader.

---

### Official Review · Reviewer_XwH1 · 2024-11-08

**Soundness:** 2
**Presentation:** 2
**Contribution:** 2
**Rating:** 3
**Confidence:** 3

**Summary:**

The paper proposed a new problem setting of knowledge preference, which attempts to unify knowledge editing and RAG. The paper proposed a benchmark for the problem and proposed a dataset synthesis method for supervised fine-tuning. Experiments show the effectiveness of the created data on the proposed benchmark.

**Strengths:**

The paper attempts to offer a unified perspective for knowledge editing and RAG: handling conflicting paramtric knowledge and external knowledge, which is interesting.

**Weaknesses:**

1. [soundness] The paper proposed that there is a hierarchy between knowledge preferences. I agree that: for knowledge editing, instruction knowledge has higher priority than parametric knowledge; for RAG, context knowledge has higher priority than parametric knowledge. But I question the hierarchy between instruction knowledge and context knowledge, and this hierarchy finds no natural applications.
- line 122 exemplifies instruction knowledge with assumptions and language requirements. The paper currently does not clearly state the rationale for this definition, and I do not agree with this definition. Requiremets and assumptions are types of constraints. It is not natural to consider them as some type of knowledge.
- Since I don't see the necessity of considering assumptions/requirements, I think the mentioned priority between instruction knowledge and context knowledge (line 147) is fabricated. There are not many occasions where user inject new knowledge (instruction knowledge) and involve retrieved knowledge (context knowledge).

2. [contribution] It remains unclear how the proposed problem helps existing research due to lack of experiements. This undermines the paper's call for addressing this new problem: people don't understand why this new problem is important.
From my understanding, addressing the proposed problem setting should alleviate the conflicts between parametric knowledge and external knowledge, as implied by Sec.1 Para.2. This should lead to improved performance on existing tasks such as multi-hop knowledge editing and RAG, as previous studies found errors due to conflicting knowledge sources. However, the current paper lacks such experimental results, and hence it remains unclear how addressing the proposed problem setting could help existing research.

3. [presentation] The methodology section lacks clarity on why creating the data is challenging, which prompts the need for the proposed data synthesis.
Supervised fine-tuning is the standard practice for a given problem, but data collection may be challenging for different problems. However, the current paper does not clearly state what is challenging about data collection and what motivates the proposed data synthesis, although they discussed the detail how they create the data.

**Questions:**

The questions correspond to the weakness section:
1. Justify why assumptions/requirements are considered as instruction knowledge.
2. What are the real applications where instruction knowledge prioritize context knowledge.
3. The effectiveness of the proposed method in improving existing multi-hop knowledge editing and RAG
4. Clarify the challenges and the motivations for the proposed method.

---

### Note · Authors · 2024-11-22

**Comment:**

Dear Reviewers,

We sincerely appreciate the time and efforts you have dedicated to reviewing this submission. We will carefully consider your comments to improve our research further, and we have decided to withdraw our paper from ICLR at this time.

Thank you once again for your constructive evaluations and feedback.

Best regards,

Authors

**Withdrawal Confirmation:**

I have read and agree with the venue's withdrawal policy on behalf of myself and my co-authors.